# LOW-RANK ROBUST GRAPH CONTRASTIVE LEARNING

## ABSTRACT

Graph Neural Networks (GNNs) have been widely used to learn node representations and with outstanding performance on various tasks such as node classification. However, noise, which inevitably exists in real-world graph data, would considerably degrade the performance of GNNs revealed by recent studies. In this work, we propose a novel and robust method, Low-Rank Robust Graph Contrastive Learning (LR-RGCL). LR-RGCL performs transductive node classification in two steps. First, a robst GCL encoder named RGCL is trained by prototypical contrastive learning with Bayesian nonparametric Prototype Learning (BPL). Next, using the robust features produced by RGCL, a novel and provable low-rank transductive classification algorithm is used to classify the unlabeled nodes in the graph. Our low-rank transductive classification algorithm is inspired by the low frequency property of the graph data and its labels, and theoretical result on the generalization of our algorithm is provided. To the best of our knowledge, our theoretical result is among the first to demonstrate the advantage of low-rank learning in transductive classification. Extensive experiments on public benchmarks demonstrate the superior performance of LR-RGCL and the robustness of the learned node representations. The code of LR-RGCL is available at https://anonymous.4open.science/r/LRR-GCL-3B3C/.

## 1 INTRODUCTION

Graph Neural Networks (GNNs) have become popular tools for node representation learning in recent years (Kipf & Welling, 2017; Bruna et al., 2014; Hamilton et al., 2017; Xu et al., 2019). Most prevailing GNNs (Kipf & Welling, 2017; Zhu & Koniusz, 2020) leverage the graph structure and obtain the representation of nodes in a graph by utilizing the features of their connected nodes. Benefiting from such propagation mechanism, node representations obtained by GNN encoders have demonstrated superior performance on various downstream tasks such as semi-supervised node classification and node clustering.

Although GNNs have achieved great success in node representation learning, many existing GNN approaches do not consider the noise in the input graph. In fact, noise inherently exists in the graph data for many real-world applications. Such noise may be present in node attributes or node labels, which forms two types of noise, attribute noise and label noise. Recent works, such as (Patrini et al., 2017), have evidenced that noisy inputs hurt the generalization capability of neural networks. Moreover, noise in a subset of the graph data can easily propagate through the graph topology to corrupt the remaining nodes in the graph data. Nodes that are corrupted by noise or falsely labeled would adversely affect the representation learning of themselves and their neighbors.

While manual data cleaning and labeling could be remedies to the consequence of noise, they are expensive processes and difficult to scale, thus not able to handle almost infinite amount of noisy data online. Therefore, it is crucial to design a robust GNN encoder that could make use of noisy training data while circumventing the adverse effect of noise. In this paper, we propose a novel and robust method termed Low-Rank Robust Graph Contrastive Learning (LR-RGCL) to improve the robustness of node representations for GNNs. We first design a new and robust GCL encoder termed RGCL. Our key observation is that there exist a subset of nodes which are confident in their class/cluster labels. Usually, such confident nodes are far away from the class/cluster boundaries, so these confident nodes are trustworthy, and noise in these nodes would not degrade the value

of these nodes in training a GNN encoder. To infer such confident nodes, we propose a novel algorithm named Bayesian nonparametric Prototype Learning (BPL). The robust prototypes as the cluster centers of the confident nodes are computed and used to train the RGCL encoder with a loss function for prototypical contrastive learning. The confident nodes are updated during each epoch of the training of the RGCL encoder, so the robust prototype representations are also updated accordingly. The robust features produced by RGCL is then used to train a novel and provable low-rank transductive node classifier.

## 1.1 CONTRIBUTIONS

Our contributions are as follows.

First, we present a novel and provable low-rank transductive node classification algorithm. Our algorithm works on the features produced by our RGCL encoder, and the algorithm is inspired by the low frequency property illustrated in Figure 1. That is, the low-rank projection of the ground truth clean labels possesses the majority of the information of the clean labels, and projection of the label noise is mostly uniform over all the eigenvectors of a kernel matrix used in classification. As a result, our algorithm only uses the low-rank part of the input features for transductive classification. We provide a novel generalization bound for the test loss on the unlabeled data, and our bound is among the first few works which exhibit the advantage of learning with low-rank features for transductive classification with the presence of noise.

Second, we propose a Robust Graph Contrastive Learning encoder termed RGCL, which is a *fully unsupervised* encoder trained on noisy graph data. The fully unsupervised RGCL encoder is trained only on the input node attributes without ground truth labels or even the ground truth class number in the training data. RGCL leverages confident nodes, which are estimated by a new algorithm termed Bayesian nonparametric Prototype Learning (BPL), to harvest noisy graph data without being compromised by the noise.

Extensive experimental results on popular graph datasets evidence the advantage of LR-RGCL over competing GNN methods for node classification on noisy graph data as well as the robustness of the RGCL encoder.

## 2 RELATED WORKS

### 2.1 GRAPH NEURAL NETWORKS

Graph neural networks (GNNs) have recently become popular tools for node representation learning. Given the difference in the convolution domain, current GNNs fall into two classes. The first class features spectral convolution (Bruna et al., 2014; Kipf & Welling, 2017), and the second class (Hamilton et al., 2017; Veličković et al., 2017; Xu et al., 2019) generates node representations by sampling and propagating features from their neighborhood. To learn node representation without node labels, contrastive learning has recently been applied to the training of GNNs (Suresh et al., 2021; Thakoor et al., 2021; Wang et al., 2022; Lee et al., 2022; Feng et al., 2022a; Zhang et al., 2023; Lin et al., 2023). Most proposed graph contrastive learning methods (Veličković et al., 2019; Sun et al., 2019; Hu et al., 2019; Jiao et al., 2020; Peng et al., 2020; You et al., 2021; Jin et al., 2021; Mo et al., 2022) create multiple views of the unlabeled input graph and maximize agreement between the node representations of these views. For example, SFA (Zhang et al., 2023) manipulates the spectrum of the node embeddings to construct augmented views in graph contrastive learning. In addition to constructing node-wise augmented views, recent works (Xu et al., 2021; Guo et al., 2022; Li et al., 2021) propose to perform contrastive learning between node representations and semantic prototype representations (Snell et al., 2017; Arik & Pfister, 2020; Allen et al., 2019; Xu et al., 2020) to encode the global semantics information.

However, as pointed out by (Dai et al., 2021), the performance of GNNs can be easily degraded by noisy training data (NT et al., 2019). Moreover, the adverse effects of noise in a subset of nodes can be exaggerated by being propagated to the remaining nodes through the network structure, exacerbating the negative impact of noise. Unlike previous GCL methods, we propose using contrastive learning to train GNN encoders that are robust to noise existing in the labels and attributes of nodes.

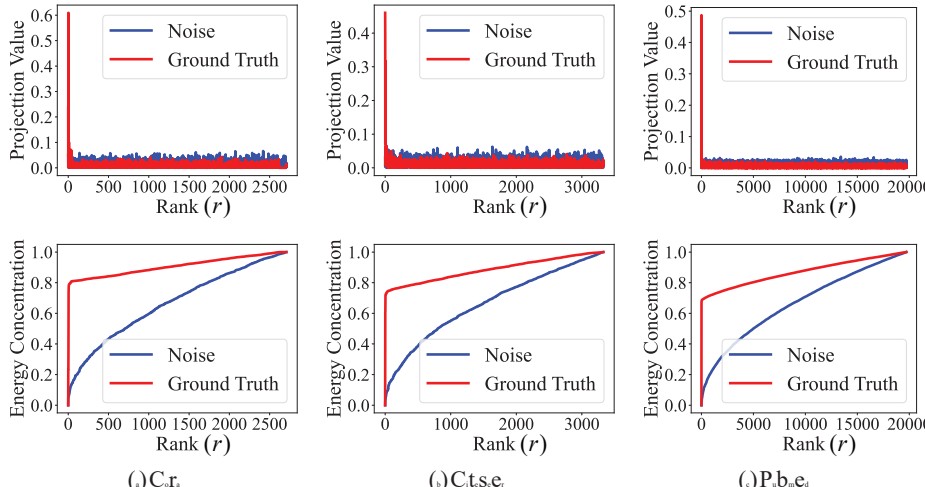

Figure 1: Eigen-projection (first row) and signal concentration ratio (second row) on Cora, Citeseer, and Pubmed. To compute the eigen-projection, we first calculate the eigenvectors $\mathbf{U}$ of the kernel gram matrix $\mathbf{K} \in \mathbb{R}^{N \times N}$ computed by a feature matrix $\mathbf{H}_{\widehat{\mathbf{A}}} \in \mathbb{R}^{N \times d}$ in Section 4.3, then the projection value is computed by $\mathbf{p} = \frac{1}{C} \sum_{c=1}^{C} \left\| \mathbf{U}^{\top} \tilde{\mathbf{Y}}^{(c)} \right\|_2^2 / \left\| \tilde{\mathbf{Y}}^{(c)} \right\|_2^2 \in \mathbb{R}^N$, where $C$ is the number of classes, and $\tilde{\mathbf{Y}} \in \{0,1\}^{N \times C}$ is the one-hot clean labels of all the nodes, $\tilde{\mathbf{Y}}^{(c)}$ is the $c$-th column of $\tilde{\mathbf{Y}}$. With the presence of label noise $\mathbf{N} \in \mathbb{R}^{N \times C}$, the observed label matrix is $\mathbf{Y} = \tilde{\mathbf{Y}} + \mathbf{N}$. The eigen-projection $\mathbf{p}_r$ for $r \in [N]$ reflects the amount of the signal projected onto the $r$-th eigenvector of $\mathbf{K}$, and the signal concentration ratio of a rank $r$ reflects the proportion of signal projected onto the top $r$ eigenvectors of $\mathbf{K}$. The signal concentration ratio for rank $r$ is computed by $\left\| \mathbf{p}^{(1:r)} \right\|_2$, where $\mathbf{p}^{(1:r)}$ contains the first $r$ elements of $\mathbf{p}$. It is observed from the red curves in the first row that the projection of the ground truth clean labels mostly concentrates on the top eigenvectors of $\mathbf{K}$. On the other hand, the projection of label noise, computed by $\frac{1}{C} \sum_{c=1}^{C} \left\| \mathbf{U}^{\top} \mathbf{N}^{(c)} \right\|_2^2 / \left\| \mathbf{Y}^{(c)} \right\|_2^2 \in \mathbb{R}^N$, is relatively uniform over all the eigenvectors, as illustrated by the blue curves in the first row. For example, by the rank $r = 0.2N$, the signal concentration ratio of $\tilde{\mathbf{Y}}$ for Cora, Citeseer, and Pubmed are $0.844$, $0.809$, and $0.784$ respectively. We refer to such property as the **low frequency property**, which suggests that we can learn a low-rank portion of the observed label $\mathbf{Y}$ which covers most information in the ground truth clean label while only learning a small portion of the label noise. Figure 3 in the supplementary further illustrates the low frequency property on more datasets.

## 2.2 Existing Methods Handing Noisy Data

Previous works (Zhang et al., 2021) have shown that deep neural networks usually generalize badly when trained on input with noise. Existing literature on robust learning mostly fall into two categories. The first category (Patrini et al., 2017; Goldberger & Ben-Reuven, 2016) mitigates the effects of noisy inputs by correcting the computation of loss function, known as loss corruption. The second category aims to select clean samples from noisy inputs for the training (Malach & Shalev-Shwartz, 2017; Jiang et al., 2018; Yu et al., 2019; Li et al., 2020; Han et al., 2018), known as sample selection. To improve the performance of GNNs on graph data with noise, NRGNN(Dai et al., 2021) first introduces a graph edge predictor to predict missing links for connecting unlabeled nodes with labeled nodes. RTGNN (Qian et al., 2022) trains a robust GNN classifier with scarce and noisy node labels. It first classifies labeled nodes into clean and noisy ones and adopts reinforcement supervision to correct noisy labels. To improve the robustness of the node classifier on the dynamic graph, GraphSS (Zhuang & Al Hasan, 2022) proposes to generalize noisy supervision as a kind of self-supervised learning method, which regards the noisy labels, including both manual-annotated labels and auto-generated labels, as one kind of self-information for each node. Different from previous works, we aim to improve the robustness of GNN encoders for node classification by applying low-rank regularization during the training of the transductive classifier.

## 3 Problem Setup

### 3.1 Notations

An attributed graph consisting of $N$ nodes is formally represented by $\mathcal{G} = (\mathcal{V}, \mathcal{E}, \mathbf{X})$, where $\mathcal{V} = \{v_1, v_2, \ldots, v_N\}$ and $\mathcal{E} \subseteq \mathcal{V} \times \mathcal{V}$ denote the set of nodes and edges respectively. $\mathbf{X} \in \mathbb{R}^{N \times D}$ are the node attributes, and the attributes of each node is in $\mathbb{R}^D$. Let $\mathbf{A} \in \{0, 1\}^{N \times N}$ be the adjacency matrix of graph $\mathcal{G}$, with $\mathbf{A}_{ij} = 1$ if and only if $(v_i, v_j) \in \mathcal{E}$. $\tilde{\mathbf{A}} = \mathbf{A} + \mathbf{I}$ denotes the adjacency matrix for a graph with self-loops added. $\tilde{\mathbf{D}}$ denotes the diagonal degree matrix of $\tilde{\mathbf{A}}$. $[n]$ denotes all the natural numbers between 1 and $N$ inclusively. $\mathcal{L}$ is a subset of $[N]$ of size $m$, and $\mathcal{U}$ is a subset of $[N] \setminus \mathcal{L}$ and $|\mathcal{U}| = u$. Let $\mathcal{V}_{\mathcal{L}}$ and $\mathcal{V}_{\mathcal{U}}$ denote the set of labeled nodes and unlabeled test nodes respectively, and $|\mathcal{V}_{\mathcal{L}}| = m$, $|\mathcal{V}_{\mathcal{U}}| = u$. Note that $m + u \leq N$, and it is not necessary that $m + u = N$ because there are usually validation nodes other than the labeled nodes and unlabeled test nodes. Let $\mathbf{u} \in \mathbb{R}^N$ be a vector, we use $[\mathbf{u}]_{\mathcal{A}}$ to denote a vector formed by elements of $\mathbf{u}$ with indices in $\mathcal{L}$ for $\mathcal{A} \subseteq [N]$ If $\mathbf{u}$ is a matrix, then $[\mathbf{u}]_{\mathcal{A}}$ denotes a submatrix formed by rows of $\mathbf{u}$ with row indices in $\mathcal{A}$. $\|\cdot\|_{\mathrm{F}}$ denotes the Frobenius norm of a matrix, and $\|\cdot\|_p$ denotes the $p$-norm of a vector.

### 3.2 Graph Convolution Network (GCN)

To learn the node representation from the attributes $\mathbf{X}$ and the graph structure $\mathbf{A}$, one simple yet effective neural network model is Graph Convolution Network (GCN). GCN is originally proposed for semi-supervised node classification, which consists of two graph convolution layers. In our work, we use GCN as the RGCL encoder to obtain node representation $\mathbf{H} \in \mathbb{R}^{N \times d}$, where the $i$-th row of $\mathbf{H}$ is the node representation of $v_i$. Thus the RGCL encoder is formulated as $\mathbf{H} = \sigma(\hat{\mathbf{A}}\sigma(\hat{\mathbf{A}}\mathbf{X}\tilde{\mathbf{W}}^{(0)})\tilde{\mathbf{W}}^{(1)})$, where $\hat{\mathbf{A}} = \tilde{\mathbf{D}}^{-1/2}\tilde{\mathbf{A}}\tilde{\mathbf{D}}^{-1/2}$. $\tilde{\mathbf{W}}^{(0)}$ and $\tilde{\mathbf{W}}^{(1)}$ are the weight matrices, and $\sigma$ is the activation function ReLU. The robust node representations produced by the RGCL encoder are used to perform transductive node classification in this paper. More details about RGCL encoder and transductive node classification are introduced in this subsection.

### 3.3 Problem Description

Noise usually exists in the input node attributes or labels of real-world graphs, which degrades the quality of the node representation obtained by common GCL encoders and affects the performance of the classifier trained on such representations. We aim to obtain node representations robust to noise in two cases, where noise is present in either the labels of $\mathcal{V}_{\mathcal{L}}$ or in the input node attributes $\mathbf{X}$. That is, we consider either noisy label or noisy input node attributes.

The goal of RGCL is to learn robust node representations by $\mathbf{H} = g(\mathbf{X}, \mathbf{A})$ such that the node representations $\{\mathbf{h}_i\}_{i=1}^N$ are robust to noise in the above two cases, where $g(\cdot)$ is the RGCL encoder. In our work, $g$ is a two-layer GCN specified in the previous subsection. The robust node representations by RGCL, $\mathbf{H} = \{\mathbf{h}_1; \mathbf{h}_2; \ldots; \mathbf{h}_N\} \in \mathbb{R}^{N \times d}$ are used for transductive node classification. In transductive node classification, a transductive classifier is trained on $\mathcal{V}_{\mathcal{L}}$, and then the classifier predicts the labels of the unlabeled test nodes in $\mathcal{V}_{\mathcal{U}}$.

## 4 Methods

### 4.1 RGCL: Robust Graph Contrastive Learning with Bayesian Nonparametric Prototype Learning (BPL)

**Preliminary of GCL.** The general node representation learning aims to train an encoder $g(\cdot)$, which is a two-layer Graph Convolution Neural Network (GCN) (Kipf & Welling, 2017), to generate discriminative node representations. In our work, we adopt contrastive learning to train the RGCL encoder $g(\cdot)$. To perform contrastive learning, two different views, $G^1 = (\mathbf{X}^1, \mathbf{A}^1)$ and $G^2 = (\mathbf{X}^2, \mathbf{A}^2)$, are generated by node dropping, edge perturbation, and attribute masking. The representation of two generated views are denoted as $\mathbf{H}^1 = g(\mathbf{X}^1, \mathbf{A}^1)$ and $\mathbf{H}^2 = g(\mathbf{X}^2, \mathbf{A}^2)$, with $\mathbf{H}_i^1$ and $\mathbf{H}_i^2$ being the $i$-th row of $\mathbf{H}^1$ and $\mathbf{H}^2$, respectively. It is preferred that the mutual information between $\mathbf{H}^1$ and $\mathbf{H}^2$ is maximized. For computational reason, its lower bound is usually used

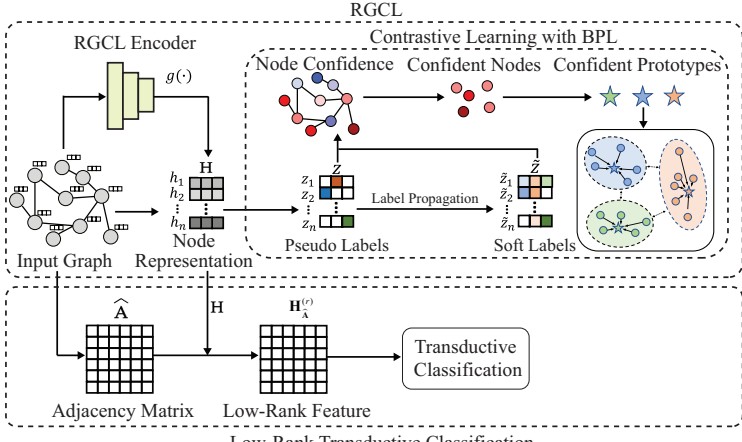

Figure 2: Illustration of the LR-RGCL framework.

as the objective for contrastive learning. We use InfoNCE (Li et al., 2021) as our node-wise contrastive loss. In addition to the node-wise contrastive learning, we also adopt prototypical contrastive learning (Li et al., 2021) to capture semantic information in the node representations, which is interpreted as maximizing the mutual information between node representation and a set of estimated cluster prototypes $\{\mathbf{c}_1, ..., \mathbf{c}_K\}$. Here $K$ is the number of cluster prototypes. The loss function for node-wise contrastive learning and prototypical contrastive learning are

$$\mathcal{L}_{node} = -\frac{1}{N} \sum_{i=1}^{N} \log \frac{s(\mathbf{H}_i^1, \mathbf{H}_i^2)}{s(\mathbf{H}_i^1, \mathbf{H}_i^2) + \sum_{j=1}^{N} s(\mathbf{H}_i^1, \mathbf{H}_j^2)}, \mathcal{L}_{proto} = -\frac{1}{N} \sum_{i=1}^{N} \log \frac{\exp(\mathbf{H}_i \cdot \mathbf{c}_k / \tau)}{\sum_{k=1}^{K} \exp(\mathbf{H}_i \cdot \mathbf{c}_k / \tau)},$$
(1)

where $s(\mathbf{H}_i^1, \mathbf{H}_i^2)$ is the cosine similarity between two node representations, $\mathbf{H}_i^1$ and $\mathbf{H}_i^2$.

**RGCL: Robust Graph Contrastive Learning.** RGCL aims to improve the robustness of node representations by prototypical contrastive learning through learning robust prototypes with confident nodes. Our key observation is that there exists a subset of nodes that are confident about their class/cluster labels because they are far away from class/cluster boundaries. We propose an effective method to infer such confident nodes. Because the RGCL encoder is completely unsupervised, it does not have access to the ground truth label or ground truth class/cluster number. Therefore, our algorithm for selection of confident nodes is based on Bayesian non-parameter style inference, and the algorithm is termed Bayesian nonparametric Prototype Learning (BPL) to be introduced next.

## 4.2 BPL: BAYESIAN NONPARAMETRIC PROTOTYPE LEARNING

We propose Bayesian nonparametric Prototype Learning which estimates robust nodes by the confidence of nodes in their labels. Intuitively, nodes more confident in their labels are less likely to be adversely affected by noise. Because RGCL is unsupervised, pseudo labels are used as the labels for such estimation. BPL, as a Bayesian nonparametric algorithm, infers the cluster prototypes by the standard Dirichlet Process Mixture Model (DPMM) under the assumption that the distribution of node representations is a mixture of Gaussians. The BPL algorithm, with details deferred to Section 4.2, produces $K$ clusters with cluster centers being the prototypes $\{\mathbf{c}_k\}_{k=1}^{K}$, where $K$ is the inferred number of prototypes.

After obtaining the cluster labels as the pseudo labels of nodes by BPL, we estimate the confidence of the nodes based on their pseudo labels and the graph structure. Let $\mathbf{z}_i$ denote the one-hot pseudo label of node $v_i$ estimated by the BPL. Label propagation (Zhang & Chen, 2018) is applied based on the adjacency matrix to get a soft pseudo label for each node. Let $\mathbf{Z} \in \mathbb{R}^{N \times K}$ be the matrix of pseudo labels with $\mathbf{z}_i$ being the $i$-th row of $\mathbf{Z}$. Let $\tilde{\mathbf{Z}}$ be the soft labels obtained by the label propagation with $\tilde{\mathbf{z}}_i$ being the $i$-th row of $\tilde{\mathbf{Z}}$. Following (Han et al., 2018), we use the cross-entropy between $\mathbf{z}_i$ and $\tilde{\mathbf{z}}_i$, denoted by $\phi(\mathbf{z}_i, \tilde{\mathbf{z}}_i)$, to identify confident nodes. Smaller cross-entropy $\phi(\mathbf{z}_i, \tilde{\mathbf{z}}_i)$ suggests that node $v_i$ is more confident about its pseudo label $\tilde{\mathbf{z}}_i$. We denote the set of confident nodes assigned to the $k$-th cluster as $\mathcal{T}_k = \{\mathbf{h}_i \mid \phi(\mathbf{z}_i, \tilde{\mathbf{z}}_i) < \gamma_k\}$, where $\gamma_k$ is a threshold for the $k$-th

---

**Algorithm 1** Training algorithm of RGCL encoder with BPL

---

**Input:** The input attribute matrix $\mathbf{X}$, adjacency matrix $\mathbf{A}$, and the training epochs $t_{\max}$.
**Output:** The parameter of RGCL encoder $g$.
1: Initialize the parameter of RGCL encoder $g$
2: **for** $t \leftarrow 1$ to $t_{\max}$ **do**
3:     Calculate node representations by $\mathbf{H} = g(\mathbf{X}, \mathbf{A})$, generate augmented views $G^1, G^2$, and calculate node representations $\mathbf{H}^1 = g(\mathbf{X}^1, \mathbf{A}^1)$ and $\mathbf{H}^2 = g(\mathbf{X}^2, \mathbf{A}^2)$.
4:     Obtain the pseudo labels of all the nodes $\mathbf{Z}$ and the number of inferred prototypes $K$ by BPL
5:     Update the confidence thresholds $\{\gamma_k\}_{k=1}^K$ and estimate the sets of confident nodes $\{\mathcal{T}_k\}_{k=1}^K$ according to Section 4.2
6:     Update confident prototypes by $\mathbf{c}_k = \frac{1}{|\mathcal{T}_k|} \sum_{\mathbf{h}_i \in \mathcal{T}_k} \mathbf{h}_i$ for all $k \in [K]$
7:     Update the parameters of RGCL encoder $g$ by one step of gradient descent on the loss $\mathcal{L}_{rep}$
8: **end for**
9: **return** The RGCL encoder $g$

---

class. The threshold $\gamma_k$ is dynamically set by $\gamma_k = 1 - \min\{\gamma_0, \gamma_0 t / t_{\max}\}$, where $t$ is the current epoch number and $t_{\max}$ is a preset number of epochs. The selected confident nodes are only used to obtain the robust prototypes, and RGCL is trained with such robust prototypes to obtain robust representations for all the nodes of the graph. $\gamma_0$ is an annealing factor which is decided by cross-validation for each dataset in practice. After acquiring the confident nodes $\{\mathcal{T}_k\}_{k=1}^K$, the prototype representations are updated by $\mathbf{c}_k = \frac{1}{|\mathcal{T}_k|} \sum_{\mathbf{h}_i \in \mathcal{T}_k} \mathbf{h}_i$ for each $k \in [K]$. With the updated cluster prototypes $\{\mathbf{c}_k\}_{k=1}^K$ in the prototypical contrastive learning loss $\mathcal{L}_{proto}$, we train the encoder $g(\cdot)$ with the overall loss function, $\mathcal{L}_{rep} = \mathcal{L}_{node} + \mathcal{L}_{proto}$. We summarize the training algorithm for the RGCL encoder in Algorithm 1. It is noted that confident nodes and robust prototypes are estimated at each epoch.

### 4.3   Low-Rank Transductive Node Classification

In this section, we introduce our novel low-rank transductive node classification algorithm using robust node representations $\mathbf{H} \in \mathbb{R}^{N \times d}$ produced by the RGCL encoder. We present strong theoretical result on the generalization bound for the test loss for our low-rank transductive algorithm with the presence of label noise.

We first give basic notations for our algorithm. Let $\mathbf{y}_i \in \mathbb{R}^C$ be the observed one-hot class label vector for node $v_i$ for all $i \in [N]$, and define $\mathbf{Y} := [\mathbf{y}_1; \mathbf{y}_2; \dots \mathbf{y}_N] \in \mathbb{R}^{N \times C}$ be the observed label matrix which may contain label noise $\mathbf{N} \in \mathbb{R}^{N \times C}$. Let $\mathbf{H}_{\widehat{\mathbf{A}}} := \widehat{\mathbf{A}}\mathbf{H}$ be the feature matrix whose rank is $r_0 \leq \min\{N, d\}$, and the singular value decomposition of $\mathbf{H}_{\widehat{\mathbf{A}}}$ is $\mathbf{H}_{\widehat{\mathbf{A}}} = \mathbf{U}\Sigma\mathbf{V}^\top$ where $\mathbf{U} \in \mathbb{R}^{n \times r_0}, \mathbf{V} \in \mathbb{R}^{d \times r_0}$ are orthogonal matrices, and $\Sigma$ is a diagonal matrix with diagonal elements $\widehat{\lambda}_1 \geq \widehat{\lambda}_2 \geq \dots \geq \widehat{\lambda}_{r_0} > 0$ being the singular values of $\mathbf{H}_{\widehat{\mathbf{A}}}$. Let $\mathbf{H}_{\widehat{\mathbf{A}}}^{(r)}$ with $r \leq r_0$ be the best rank $r$-approximation to $\mathbf{H}_{\widehat{\mathbf{A}}}$. Let $\mathbf{K} = \mathbf{H}_{\widehat{\mathbf{A}}}\mathbf{H}_{\widehat{\mathbf{A}}}^\top \in \mathbb{R}^{N \times N}$ be the kernel gram matrix of the low-rank features $\mathbf{H}_{\widehat{\mathbf{A}}}^{(r)}$, and $\mathbf{K}^{(r)} = \mathbf{H}_{\widehat{\mathbf{A}}}^{(r)} \left(\mathbf{H}_{\widehat{\mathbf{A}}}^{(r)}\right)^\top$ be the gram matrix using the low-rank features $\mathbf{H}_{\widehat{\mathbf{A}}}^{(r)}$. We use $\mathbf{U}^{(r)} \in \mathbb{R}^{N \times r}$ with $r \leq r_0$ to denote the top-$r$ eigenvectors of $\mathbf{K}$, which are the first $r$ columns of $\mathbf{U}$.

**Motivation of Low-Rank Transductive Classification.** Let $\tilde{\mathbf{Y}} \in \mathbb{R}^{N \times C}$ be the ground truth clean label matrix without noise. By the low frequency property illustrated in Figure 1, the projection of $\tilde{\mathbf{Y}}$ on the top $r$ eigenvectors of $\mathbf{K}$ with a small rank $r$, such as $r = 0.2N$, covers the majority of the information in $\tilde{\mathbf{Y}}$. On the other hand, the projection of the label noise $\mathbf{N}$ are distributed mostly uniform across all the eigenvectors. This observation motivates a low-rank transductive classification method where only the low-rank part of the feature matrix $\mathbf{H}_{\widehat{\mathbf{A}}}$ is used in classification. This is because the low-rank part of the feature matrix, which is $\mathbf{H}_{\widehat{\mathbf{A}}}^{(r)}$, suffices for learning the dominant information in the ground truth label $\tilde{\mathbf{Y}}$ while learning only a small portion of the label noise.

Let $\mathbf{F}(\mathbf{W}, r) = \mathbf{H}_{\widehat{\mathbf{A}}}^{(r)}\mathbf{W}$ with $\mathbf{W} \in \mathbb{R}^{d \times C}$ being the weight matrix for the transductive classifier. Our transductive classifier uses $\text{softmax}(\mathbf{F}(\mathbf{W}, r)) \in \mathbb{R}^{n \times C}$ for prediction of the labels of the test nodes using the low-rank part of the features, $\mathbf{H}_{\widehat{\mathbf{A}}}^{(r)}$. We train the transductive classifier by minimizing the regular cross-entropy on the labeled nodes via

$$\min_{\mathbf{W}} L(\mathbf{W}) = \frac{1}{m} \sum_{v_i \in \mathcal{V}_{\mathcal{L}}} \text{KL}\left(\mathbf{y}_i, \left[\text{softmax}\left(\mathbf{H}_{\widehat{\mathbf{A}}}^{(r)}\mathbf{W}\right)\right]_i\right), \tag{2}$$

where KL is the KL divergence between the label $\mathbf{y}_i$ and the softmax of the classifier output at node $v_i$. We use a regular gradient descent to optimize (2) with a learning rate $\eta \in (0, \frac{1}{\widehat{\lambda}_1})$. We define a matrix $\mathbf{Y}^{\perp} \in \mathbb{R}^{N \times C}$ as the orthogonal projection of $\mathbf{Y}$ onto the top-$r$ eigenvectors of $\mathbf{K}$, that is, $\mathbf{Y}^{\perp} = \mathbf{U}^{(r)}\left(\mathbf{U}^{(r)}\right)^{\top}\mathbf{Y}$. $\mathbf{W}$ is initialized by $\mathbf{W}^{(0)} = \mathbf{0}$, and at the $t$-th iteration of gradient descent for $t \geq 1$, $\mathbf{W}$ is updated by $\mathbf{W}^{(t)} = \mathbf{W}^{(t-1)} - \eta \nabla_{\mathbf{W}} L(\mathbf{W})|_{\mathbf{W}=\mathbf{W}^{(t-1)}}$.

Define $\mathbf{F}(\mathbf{W}, r, t) := \mathbf{H}_{\widehat{\mathbf{A}}}^{(r)}\mathbf{W}^{(t)}$ as the output of the classifier after the $t$-th iteration of gradient descent for $t \geq 1$. We have the following theoretical result on the loss of the unlabeled test nodes $\mathcal{V}_{\mathcal{U}}$ measured by the gap between $\mathbf{F}(\mathbf{W}, r, t)$ and $\bar{\mathbf{Y}}(r)$ when using the low-rank feature $\mathbf{H}_{\widehat{\mathbf{A}}}^{(r)}$ with $r \in [r_0]$.

**Theorem 4.1.** Let $m \geq cN$ for a constant $c \in (0, 1)$, and $r \in [r_0]$. Assume that a set $\mathcal{L}$ with $|\mathcal{L}| = m$ is sampled uniformly without replacement from $[N]$, and a set $\mathcal{U}$ with $|\mathcal{U}| = u$ are sampled uniformly without replacement from $[N] \backslash \mathcal{L}$ and $m + u \leq N$. Then for every $x > 0$, with probability at least $1 - \exp(-x)$, after the $t$-th iteration of gradient descent for all $t \geq 1$, we have

$$\mathcal{U}_{\text{test}}(t) := \frac{1}{u}\left\|\left[\mathbf{F}(\mathbf{W}, r, t) - \bar{\mathbf{Y}}(r)\right]_{\mathcal{U}}\right\|_{\text{F}}^2 \leq \frac{1 + 1/c}{m}\left(1 - \eta\widehat{\lambda}_r^2\right)^{2t}\|\mathbf{Y}\|_{\text{F}}^2 + c_1 c_3 r\left(\frac{1}{u} + \frac{1}{m}\right) + \frac{c_2 x}{u}, \tag{3}$$

where $c_1, c_2, c_3$ are positive numbers depending on $\mathbf{U}$, $\left\{\widehat{\lambda}_i\right\}_{i=1}^r$, and $\tau_0$ with $\tau_0^2 = \max_{i \in [N]} \mathbf{K}_{ii}$.

This theorem is proved in Section A of the supplementary. It is noted that $\frac{1}{u}\left\|\left[\mathbf{F}(\mathbf{W}, r, t) - \bar{\mathbf{Y}}(r)\right]_{\mathcal{U}}\right\|_{\text{F}}^2$ is the test loss of the unlabeled nodes measured by the distance between the classifier output $\mathbf{F}(\mathbf{W}, r, t)$ and $\bar{\mathbf{Y}}(r)$. We note that $\bar{\mathbf{Y}}(r) = \mathbf{U}^{(r)}\left(\mathbf{U}^{(r)}\right)^{\top}\tilde{\mathbf{Y}} + \mathbf{U}^{(r)}\left(\mathbf{U}^{(r)}\right)^{\top}\mathbf{N}$ is the sum of the rank-$r$ projection of the clean label $\tilde{\mathbf{Y}}$ and the rank-$r$ projection of the label noise $\mathbf{N}$. As discussed above and in the description of the low frequency property in Figure 1, the low-rank projection of $\tilde{\mathbf{Y}}$ keeps the majority of the information in the clean label while only admitting a small portion of the label noise. As a result, a small test loss $\mathcal{U}_{\text{test}}(t)$ on the LHS of the bound (9) indicates a better approximation to the clean label of the unlabeled test nodes. On the other hand, with sufficient training via a large $t$, we have $\mathcal{U}_{\text{test}}(t) \leq c_1 c_3 r\left(\frac{1}{u} + \frac{1}{m}\right) + \frac{c_2 x}{u} + \varepsilon(t)$ with $\varepsilon(t) \xrightarrow{t \to \infty} 0$. This indicates that a relatively smaller rank $r$ indicates better approximation to $\bar{\mathbf{Y}}(r)$. On the other hand, the rank $r$ should not be too small so that $\mathbf{Y}(r)$ can contain enough information from the clean labels. In Table 6 of our experimental results, it is observed that the performance of our low-rank transductive classifier is consistent with rank $0.1 \min\{N, d\} \leq r \leq 0.2 \min\{N, d\}$. We set $r = 0.2 \min\{N, d\}$ for all the experiments throughout this paper. The overall framework of LR-RGCL is illustrated in Figure 2.

## 5 EXPERIMENTS

### 5.1 EXPERIMENTAL SETTINGS

In our experiment, we adopt eight widely used graph benchmark datasets, namely Cora, Citeseer, PubMed (Sen et al., 2008), Coauthor CS, ogbn-arxiv (Hu et al., 2020), Wiki-CS (Mernyei & Cangea, 2020), Amazon-Computers, and Amazon-Photos (Shchur et al., 2018) for the evaluation in node classification. Details of the datasets are deferred in Section C.1 of the supplementary. Due to the fact that all public benchmark graph datasets do not come with corrupted labels or attribute noise, we manually inject noise into public datasets to evaluate our algorithm. We follow the commonly used

Table 1: Performance comparison for node classification on Cora, Citeseer, PubMed, and Wiki-CS with asymmetric label noise, symmetric label noise, and attribute noise.

| Dataset | Methods | Noise Level | | | | | | | | | |
| | | 0 | 40 | | | 60 | | | 80 | | |
| | | - | Asymmetric | Symmetric | Attribute | Asymmetric | Symmetric | Attribute | Asymmetric | Symmetric | Attribute |
|---|---|---|---|---|---|---|---|---|---|---|---|
| Cora | GCN | 0.815±0.005 | 0.547±0.015 | 0.636±0.007 | 0.639±0.008 | 0.405±0.014 | 0.517±0.010 | 0.439±0.012 | 0.265±0.012 | 0.354±0.014 | 0.317±0.013 |
| | S$^2$GC | 0.835±0.002 | 0.569±0.007 | 0.664±0.007 | 0.661±0.007 | 0.422±0.010 | 0.535±0.010 | 0.454±0.011 | 0.279±0.014 | 0.366±0.014 | 0.320±0.013 |
| | GCE | 0.819±0.004 | 0.573±0.011 | 0.652±0.008 | 0.650±0.014 | 0.449±0.011 | 0.509±0.011 | 0.445±0.015 | 0.280±0.013 | 0.353±0.013 | 0.325±0.015 |
| | UnionNET | 0.820±0.006 | 0.569±0.014 | 0.664±0.007 | 0.653±0.012 | 0.452±0.010 | 0.541±0.010 | 0.450±0.009 | 0.283±0.014 | 0.370±0.011 | 0.320±0.012 |
| | NRGNN | 0.822±0.006 | 0.571±0.019 | 0.676±0.007 | 0.645±0.012 | 0.470±0.014 | 0.548±0.014 | 0.451±0.011 | 0.282±0.022 | 0.373±0.012 | 0.326±0.010 |
| | RTGNN | 0.828±0.003 | 0.570±0.010 | 0.682±0.008 | 0.678±0.011 | 0.474±0.011 | 0.555±0.010 | 0.457±0.009 | 0.280±0.011 | 0.386±0.014 | 0.342±0.016 |
| | SUGRL | 0.834±0.005 | 0.564±0.011 | 0.674±0.012 | 0.675±0.009 | 0.468±0.011 | 0.552±0.011 | 0.452±0.012 | 0.280±0.012 | 0.381±0.012 | 0.338±0.014 |
| | MERIT | 0.831±0.005 | 0.560±0.008 | 0.670±0.008 | 0.671±0.009 | 0.467±0.013 | 0.547±0.013 | 0.450±0.014 | 0.277±0.013 | 0.385±0.013 | 0.335±0.009 |
| | ARIEL | 0.843±0.004 | 0.573±0.013 | 0.681±0.010 | 0.675±0.009 | 0.471±0.012 | 0.553±0.012 | 0.455±0.014 | 0.284±0.014 | 0.389±0.013 | 0.343±0.013 |
| | SFA | 0.839±0.010 | 0.564±0.011 | 0.677±0.013 | 0.676±0.015 | 0.473±0.014 | 0.549±0.014 | 0.457±0.014 | 0.282±0.016 | 0.389±0.013 | 0.344±0.017 |
| | Sel-Cl | 0.828±0.002 | 0.570±0.010 | 0.685±0.012 | 0.676±0.009 | 0.472±0.013 | 0.554±0.014 | 0.455±0.011 | 0.282±0.017 | 0.389±0.013 | 0.341±0.015 |
| | Jo-SRC | 0.825±0.005 | 0.571±0.006 | 0.684±0.013 | 0.679±0.007 | 0.473±0.011 | 0.556±0.008 | 0.458±0.012 | 0.285±0.013 | 0.387±0.018 | 0.345±0.018 |
| | GRAND+ | 0.858±0.006 | 0.570±0.009 | 0.682±0.007 | 0.678±0.011 | 0.472±0.010 | 0.554±0.008 | 0.456±0.012 | 0.284±0.015 | 0.387±0.015 | 0.345±0.013 |
| | RGCL | 0.854±0.006 | 0.584±0.009 | 0.704±0.007 | 0.690±0.010 | 0.484±0.013 | 0.577±0.013 | 0.469±0.013 | 0.295±0.012 | 0.407±0.012 | 0.356±0.011 |
| | LR-RGCL | **0.858±0.006** | **0.589±0.011** | **0.713±0.007** | **0.695±0.011** | **0.492±0.011** | **0.587±0.013** | **0.477±0.012** | **0.306±0.012** | **0.419±0.012** | **0.363±0.011** |
| Citeseer | GCN | 0.703±0.005 | 0.475±0.023 | 0.501±0.013 | 0.529±0.009 | 0.351±0.014 | 0.341±0.014 | 0.372±0.011 | 0.291±0.022 | 0.281±0.019 | 0.290±0.014 |
| | S$^2$GC | 0.736±0.005 | 0.488±0.013 | 0.528±0.013 | 0.553±0.008 | 0.363±0.012 | 0.367±0.014 | 0.390±0.013 | 0.304±0.024 | 0.284±0.019 | 0.288±0.011 |
| | GCE | 0.705±0.004 | 0.490±0.016 | 0.512±0.014 | 0.540±0.014 | 0.362±0.015 | 0.352±0.010 | 0.381±0.009 | 0.309±0.012 | 0.285±0.014 | 0.285±0.011 |
| | UnionNET | 0.706±0.006 | 0.499±0.015 | 0.547±0.014 | 0.545±0.013 | 0.379±0.013 | 0.399±0.013 | 0.379±0.012 | 0.322±0.021 | 0.302±0.013 | 0.284±0.009 |
| | NRGNN | 0.710±0.008 | 0.498±0.015 | 0.546±0.015 | 0.538±0.011 | 0.382±0.016 | 0.412±0.016 | 0.377±0.012 | 0.336±0.021 | 0.309±0.018 | 0.284±0.009 |
| | RTGNN | 0.746±0.008 | 0.498±0.007 | 0.556±0.007 | 0.550±0.012 | 0.392±0.010 | 0.424±0.013 | 0.390±0.014 | 0.348±0.017 | 0.308±0.016 | 0.302±0.011 |
| | SUGRL | 0.730±0.005 | 0.493±0.011 | 0.541±0.011 | 0.544±0.010 | 0.376±0.009 | 0.421±0.009 | 0.388±0.009 | 0.339±0.010 | 0.305±0.010 | 0.300±0.009 |
| | MERIT | 0.740±0.007 | 0.496±0.012 | 0.536±0.012 | 0.542±0.010 | 0.383±0.011 | 0.425±0.011 | 0.387±0.008 | 0.344±0.014 | 0.301±0.014 | 0.295±0.009 |
| | SFA | 0.740±0.011 | 0.502±0.014 | 0.532±0.015 | 0.547±0.013 | 0.390±0.014 | 0.433±0.014 | 0.389±0.012 | 0.347±0.016 | 0.312±0.015 | 0.299±0.013 |
| | ARIEL | 0.727±0.007 | 0.500±0.008 | 0.550±0.013 | 0.548±0.008 | 0.391±0.009 | 0.427±0.012 | 0.389±0.014 | 0.349±0.014 | 0.307±0.013 | 0.299±0.013 |
| | Sel-Cl | 0.725±0.008 | 0.499±0.012 | 0.551±0.010 | 0.549±0.008 | 0.389±0.011 | 0.426±0.008 | 0.391±0.012 | 0.350±0.018 | 0.310±0.015 | 0.300±0.017 |
| | Jo-SRC | 0.730±0.005 | 0.500±0.013 | 0.555±0.011 | 0.551±0.011 | 0.394±0.013 | 0.425±0.013 | 0.393±0.013 | 0.351±0.013 | 0.305±0.018 | 0.303±0.013 |
| | GRAND+ | 0.756±0.004 | 0.497±0.010 | 0.553±0.012 | 0.552±0.011 | 0.390±0.013 | 0.422±0.013 | 0.387±0.013 | 0.348±0.013 | 0.309±0.014 | 0.302±0.012 |
| | RGCL | 0.748±0.009 | 0.510±0.013 | 0.574±0.013 | 0.562±0.007 | 0.403±0.014 | 0.445±0.014 | 0.399±0.012 | 0.359±0.012 | 0.327±0.014 | 0.312±0.010 |
| | LR-RGCL | **0.757±0.010** | **0.520±0.013** | **0.581±0.013** | **0.570±0.007** | **0.414±0.014** | **0.455±0.014** | **0.406±0.012** | **0.369±0.012** | **0.335±0.014** | **0.318±0.010** |
| PubMed | GCN | 0.790±0.007 | 0.584±0.022 | 0.574±0.012 | 0.595±0.012 | 0.405±0.025 | 0.386±0.011 | 0.488±0.013 | 0.305±0.022 | 0.295±0.013 | 0.423±0.013 |
| | S$^2$GC | 0.802±0.005 | 0.585±0.023 | 0.589±0.013 | 0.610±0.009 | 0.421±0.030 | 0.401±0.014 | 0.497±0.012 | 0.310±0.039 | 0.290±0.019 | 0.431±0.010 |
| | GCE | 0.792±0.009 | 0.589±0.018 | 0.581±0.011 | 0.590±0.014 | 0.430±0.012 | 0.399±0.012 | 0.491±0.010 | 0.311±0.021 | 0.301±0.011 | 0.424±0.012 |
| | UnionNET | 0.793±0.008 | 0.603±0.020 | 0.620±0.012 | 0.592±0.012 | 0.445±0.022 | 0.424±0.013 | 0.489±0.015 | 0.313±0.025 | 0.327±0.015 | 0.435±0.009 |
| | NRGNN | 0.797±0.008 | 0.602±0.022 | 0.618±0.013 | 0.603±0.008 | 0.443±0.012 | 0.434±0.012 | 0.499±0.009 | 0.330±0.023 | 0.325±0.013 | 0.433±0.011 |
| | RTGNN | 0.797±0.004 | 0.610±0.008 | 0.622±0.010 | 0.614±0.012 | 0.455±0.010 | 0.455±0.011 | 0.501±0.011 | 0.335±0.013 | 0.338±0.017 | 0.452±0.013 |
| | SUGRL | 0.819±0.005 | 0.603±0.013 | 0.615±0.013 | 0.615±0.010 | 0.445±0.011 | 0.441±0.011 | 0.501±0.007 | 0.321±0.009 | 0.321±0.009 | 0.446±0.010 |
| | MERIT | 0.801±0.004 | 0.593±0.011 | 0.612±0.011 | 0.613±0.011 | 0.447±0.012 | 0.443±0.012 | 0.497±0.009 | 0.328±0.011 | 0.323±0.011 | 0.445±0.009 |
| | ARIEL | 0.800±0.003 | 0.610±0.013 | 0.622±0.010 | 0.615±0.011 | 0.453±0.012 | 0.453±0.012 | 0.502±0.014 | 0.331±0.014 | 0.336±0.018 | 0.457±0.013 |
| | SFA | 0.804±0.010 | 0.596±0.011 | 0.615±0.011 | 0.609±0.011 | 0.447±0.014 | 0.446±0.017 | 0.499±0.014 | 0.330±0.011 | 0.327±0.011 | 0.447±0.014 |
| | Sel-Cl | 0.799±0.005 | 0.605±0.014 | 0.625±0.012 | 0.614±0.012 | 0.455±0.014 | 0.449±0.010 | 0.502±0.008 | 0.334±0.021 | 0.332±0.014 | 0.456±0.014 |
| | Jo-SRC | 0.801±0.005 | 0.613±0.010 | 0.624±0.013 | 0.617±0.013 | 0.453±0.008 | 0.455±0.013 | 0.504±0.013 | 0.330±0.015 | 0.334±0.018 | 0.459±0.018 |
| | GRAND+ | 0.845±0.006 | 0.610±0.011 | 0.624±0.013 | 0.617±0.013 | 0.453±0.008 | 0.453±0.011 | 0.503±0.010 | 0.331±0.014 | 0.337±0.013 | 0.458±0.014 |
| | RGCL | 0.840±0.007 | 0.631±0.014 | 0.640±0.010 | 0.633±0.011 | 0.472±0.011 | 0.477±0.010 | 0.520±0.011 | 0.350±0.014 | 0.355±0.013 | 0.476±0.011 |
| | LR-RGCL | **0.845±0.009** | **0.637±0.014** | **0.645±0.015** | **0.637±0.011** | **0.479±0.014** | **0.484±0.015** | **0.526±0.011** | **0.356±0.011** | **0.360±0.012** | **0.482±0.014** |
| Coauthor-CS | GCN | 0.918±0.001 | 0.645±0.009 | 0.656±0.006 | 0.702±0.010 | 0.511±0.013 | 0.501±0.009 | 0.531±0.010 | 0.429±0.022 | 0.389±0.011 | 0.415±0.013 |
| | S$^2$GC | 0.918±0.001 | 0.657±0.012 | 0.663±0.006 | 0.713±0.010 | 0.516±0.013 | 0.514±0.009 | 0.556±0.009 | 0.437±0.020 | 0.396±0.010 | 0.422±0.012 |
| | GCE | 0.922±0.003 | 0.662±0.017 | 0.659±0.007 | 0.705±0.014 | 0.515±0.016 | 0.502±0.007 | 0.539±0.009 | 0.443±0.017 | 0.389±0.012 | 0.412±0.011 |
| | UnionNET | 0.918±0.002 | 0.669±0.023 | 0.671±0.013 | 0.706±0.012 | 0.525±0.011 | 0.529±0.011 | 0.540±0.012 | 0.458±0.015 | 0.401±0.011 | 0.420±0.007 |
| | NRGNN | 0.919±0.002 | 0.678±0.014 | 0.689±0.009 | 0.705±0.012 | 0.545±0.021 | 0.556±0.011 | 0.546±0.011 | 0.461±0.012 | 0.410±0.012 | 0.417±0.007 |
| | RTGNN | 0.920±0.005 | 0.678±0.012 | 0.691±0.009 | 0.712±0.008 | 0.559±0.010 | 0.569±0.011 | 0.560±0.008 | 0.455±0.015 | 0.415±0.015 | 0.412±0.014 |
| | SUGRL | 0.922±0.004 | 0.675±0.010 | 0.695±0.010 | 0.714±0.006 | 0.550±0.011 | 0.560±0.011 | 0.561±0.007 | 0.449±0.011 | 0.411±0.011 | 0.429±0.008 |
| | MERIT | 0.924±0.004 | 0.679±0.011 | 0.689±0.008 | 0.709±0.005 | 0.552±0.014 | 0.562±0.014 | 0.562±0.011 | 0.452±0.013 | 0.403±0.013 | 0.426±0.005 |
| | ARIEL | 0.925±0.004 | 0.682±0.011 | 0.699±0.009 | 0.712±0.005 | 0.555±0.011 | 0.566±0.011 | 0.556±0.011 | 0.454±0.014 | 0.415±0.019 | 0.427±0.013 |
| | SFA | 0.925±0.009 | 0.682±0.011 | 0.690±0.012 | 0.715±0.012 | 0.555±0.015 | 0.567±0.014 | 0.565±0.013 | 0.458±0.013 | 0.402±0.013 | 0.429±0.015 |
| | Sel-Cl | 0.922±0.008 | 0.684±0.009 | 0.694±0.012 | 0.714±0.010 | 0.557±0.013 | 0.568±0.013 | 0.566±0.010 | 0.457±0.013 | 0.412±0.017 | 0.425±0.009 |
| | Jo-SRC | 0.921±0.005 | 0.684±0.011 | 0.695±0.004 | 0.709±0.007 | 0.560±0.011 | 0.566±0.011 | 0.561±0.009 | 0.456±0.011 | 0.410±0.018 | 0.428±0.010 |
| | GRAND+ | 0.927±0.004 | 0.682±0.011 | 0.693±0.006 | 0.715±0.008 | 0.554±0.008 | 0.568±0.013 | 0.557±0.011 | 0.455±0.012 | 0.416±0.013 | 0.428±0.011 |
| | RGCL | 0.929±0.006 | 0.694±0.013 | 0.718±0.008 | 0.733±0.009 | 0.570±0.014 | 0.587±0.011 | 0.585±0.012 | 0.465±0.015 | 0.434±0.015 | 0.444±0.012 |
| | LR-RGCL | **0.933±0.006** | **0.699±0.015** | **0.721±0.012** | **0.742±0.015** | **0.575±0.014** | **0.595±0.018** | **0.588±0.015** | **0.469±0.015** | **0.438±0.015** | **0.453±0.017** |
| ogbn-arxiv | GCN | 0.717±0.003 | 0.401±0.014 | 0.421±0.014 | 0.478±0.010 | 0.336±0.011 | 0.346±0.021 | 0.339±0.012 | 0.286±0.022 | 0.256±0.010 | 0.294±0.013 |
| | S$^2$GC | 0.712±0.003 | 0.417±0.017 | 0.429±0.014 | 0.492±0.010 | 0.344±0.016 | 0.353±0.031 | 0.343±0.009 | 0.297±0.023 | 0.266±0.013 | 0.284±0.012 |
| | GCE | 0.720±0.004 | 0.410±0.018 | 0.428±0.008 | 0.480±0.014 | 0.348±0.019 | 0.344±0.019 | 0.342±0.015 | 0.310±0.014 | 0.260±0.011 | 0.275±0.015 |
| | UnionNET | 0.724±0.006 | 0.429±0.021 | 0.449±0.007 | 0.485±0.012 | 0.362±0.018 | 0.367±0.008 | 0.340±0.009 | 0.332±0.019 | 0.269±0.013 | 0.280±0.012 |
| | NRGNN | 0.721±0.006 | 0.449±0.014 | 0.466±0.009 | 0.485±0.012 | 0.371±0.020 | 0.379±0.008 | 0.342±0.011 | 0.330±0.018 | 0.271±0.018 | 0.300±0.010 |
| | RTGNN | 0.718±0.004 | 0.443±0.012 | 0.464±0.012 | 0.484±0.014 | 0.380±0.011 | 0.384±0.013 | 0.340±0.017 | 0.335±0.011 | 0.285±0.015 | 0.301±0.006 |
| | SUGRL | 0.693±0.002 | 0.439±0.010 | 0.467±0.010 | 0.480±0.012 | 0.365±0.013 | 0.385±0.011 | 0.341±0.009 | 0.327±0.011 | 0.275±0.011 | 0.295±0.011 |
| | MERIT | 0.717±0.004 | 0.442±0.009 | 0.463±0.009 | 0.483±0.010 | 0.368±0.011 | 0.381±0.011 | 0.341±0.012 | 0.324±0.012 | 0.272±0.010 | 0.304±0.009 |
| | ARIEL | 0.717±0.004 | 0.448±0.013 | 0.471±0.013 | 0.482±0.011 | 0.379±0.014 | 0.384±0.015 | 0.342±0.015 | 0.334±0.014 | 0.280±0.013 | 0.300±0.010 |
| | SFA | 0.718±0.009 | 0.445±0.012 | 0.463±0.013 | 0.486±0.012 | 0.368±0.011 | 0.378±0.014 | 0.338±0.015 | 0.325±0.014 | 0.273±0.012 | 0.302±0.013 |
| | Sel-Cl | 0.719±0.002 | 0.447±0.007 | 0.469±0.007 | 0.486±0.010 | 0.375±0.008 | 0.389±0.025 | 0.344±0.013 | 0.331±0.008 | 0.284±0.019 | 0.304±0.012 |
| | Jo-SRC | 0.715±0.005 | 0.445±0.011 | 0.466±0.009 | 0.481±0.010 | 0.377±0.013 | 0.387±0.013 | 0.340±0.013 | 0.333±0.013 | 0.282±0.018 | 0.297±0.009 |
| | GRAND+ | 0.725±0.004 | 0.445±0.008 | 0.466±0.011 | 0.481±0.011 | 0.378±0.010 | 0.385±0.012 | 0.344±0.010 | 0.332±0.010 | 0.282±0.016 | 0.303±0.009 |
| | RGCL | 0.727±0.005 | 0.468±0.013 | 0.487±0.006 | 0.502±0.010 | 0.400±0.014 | 0.407±0.009 | 0.359±0.011 | 0.352±0.012 | 0.303±0.013 | 0.330±0.012 |
| | LR-RGCL | **0.728±0.006** | **0.472±0.013** | **0.492±0.011** | **0.508±0.014** | **0.405±0.014** | **0.411±0.012** | **0.405±0.012** | **0.359±0.015** | **0.307±0.013** | **0.335±0.013** |

label noise generation methods from the existing work (Han et al., 2020; Dai et al., 2022; Qian et al., 2022) to inject label noise. We generate noisy labels over all classes in two types: (1) Symmetric, where nodes from each class is flipped to other classes with a uniform random probability; (2) Asymmetric, where mislabeling only occurs between similar classes. The percentage of nodes with flipped labels is defined as the label noise level in our experiments. To evaluate the performance of our method with attribute noise, we randomly shuffle a certain percentage of input attributes for each node following (Ding et al., 2022). The percentage of shuffled attributes is defined as the attribute noise level in our experiments.

## 5.2 NODE CLASSIFICATION

**Compared Methods.** We compare RGCL against semi-supervised node representation learning methods, GCN (Kipf & Welling, 2017), GCE (Zhang & Sabuncu, 2018), S$^2$GC (Zhu & Koniusz, 2020), and GRAND+ (Feng et al., 2022b). Furthermore, we include two baseline methods for

node classification with label noise, which are NRGNN (Dai et al., 2021) and RTGNN (Qian et al., 2022). We also compare RGCL against state-of-the-art GCL methods, including GraphCL (You et al., 2020), MERIT (Jin et al., 2021), SUGRL (Mo et al., 2022), Jo-SRC (Yao et al., 2021), Sel-CL (Li et al., 2022), and SFA (Zhang et al., 2023). Among the compared contrastive learning methods, Jo-SRC and Sel-CL are specifically designed for robust learning. SFA is a method that aims to improve the performance of contrastive learning with spectral augmentation. We include details of compared methods in Section C.2 of the supplementary.

**Experimental Results.** We first compare LR-RGCL against competing methods for semi-supervised or transductive node classification on input with two types of label noise. To show the robustness of LR-RGCL against label noise, we perform the experiments on graphs injected with different levels of label noise ranging from $40\%$ to $80\%$ with a step of $20\%$. We follow the widely used semi-supervised setting (Kipf & Welling, 2017) for node classification. In LR-RGCL, we train a transductive classifier for node classification. Previous GCL methods, including MERIT, SUGRL, and SFA, train a linear layer for inductive classification on top of the node representations learned by contrastive learning without using test data in training. Because LR-RGCL is a transductive classifier, for fair comparisons, we also train the compared GCL baselines with the same transductive classifier as that for LR-RGCL and a two-layer GCN transductive classifier. The results with different types of classifiers are deferred in Section D.3 of the supplementary. For all the baselines in our experiments which perform inductive classification when predicting tbe labels, we report their best results among using their original inductive classifier and two types of transductive classifiers: the same transductive classifier as that for LR-RGCL and a two-layer GCN transductive classifier.

Results on Cora, Citeseer, PubMed, Coauthor-CS, and ogbn-arxiv are shown in Table 1, where we report the means of the accuracy of 10 runs and the standard deviation. Results on Wiki-CS, Amazon-Computers, and Amazon-Photos are deferred in Section D.2 of the supplementary. It is observed from the results that LR-RGCL outperforms all the baselines. By selecting confident nodes and computing robust prototypes using BEC, LR-RGCL outperforms all the baselines by an even larger margin with a larger label noise level. In addition, we compare LR-RGCL with baselines for noisy input with attribute noise levels ranging from $40\%$ to $80\%$ with a step of $20\%$. Results on Cora, Citeseer, and Coauthor CS are shown in Table 4 in the supplementary, where we report the means of the accuracy of 10 runs and the standard deviation. The results clearly show that LR-RGCL is more robust to attribute noise compared to all the baselines for different noise levels.

RGCL in all the result tables performs transductive node classification by using the full-rank feature in LR-RGCL, that is, we set $r = r_0$ in (2). It can be observed that RGCL usually achieves the second best result across all the noise levels. LR-RGCL always performs better than RGCL, evidencing the advantage of the proposed low-rank transductive learning algorithm.

**Additional Results and Ablation Studies** We compare the training time of LR-RGCL with competing baselines in Table 7 of the supplementary. We also perform ablation study on the value of rank $r$ in the optimization problem (2) for our low-rank transductive classifier. It is observed from Table 6 of the supplementary that the performance of our low-rank classifier is consistently close to the best performance among all the choices of the rank when $r$ is between $0.1 \min \{N, d\}$ and $0.2 \min \{N, d\}$. In order to visualize the robustness of the RGCL encoder, the confidence score $\phi(\mathbf{z}_i, \tilde{\mathbf{z}}_i)$ described in Section 4.2 of all the nodes of the Citeseer data set in the embedding space of the learned node representations is illustrated in Figure 4 of the supplementary.

## 6 CONCLUSIONS

In this paper, we propose a novel transductive node classification method for noisy graph data termed Low-Rank Robust Graph Contrastive Learning (LR-RGCL). LR-RGCL trains a robust GCL encoder to learn robust node representations. It then uses the low-rank features inspired by sharp generalization bound for transductive learning to perform transductive node classification. We evaluate the performance of LR-RGCL with comparison to competing baselines on semi-supervised or transductive node classification, where graph data are corrupted with noise in either the labels for the node attributes. Extensive experimental results demonstrate that LR-RGCL generates more robust node representations with better performance than the current state-of-the-art node representation learning methods.

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

## A    THEORETICAL RESULTS

First, recall that the optimization problem of the low-rank transductive classification is

$$
\min_{\mathbf{W}} \frac{1}{m} \sum_{v_i \in \mathcal{V}_\mathcal{L}} \mathrm{KL}\left(\mathbf{y}_i, \left[\mathrm{softmax}\left(\mathbf{H}_{\widehat{\mathbf{A}}}^{(r)}\mathbf{W}\right)\right]_i\right).
$$

We then present the proof of Theorem 4.1.

**Proof of Theorem 4.1.**  It can be verified that at the $t$-th iteration of gradient descent for $t \geq 1$, we have

$$
\mathbf{W}^{(t)} = \mathbf{W}^{(t-1)} - \eta\left(\mathbf{H}_{\widehat{\mathbf{A}}}^{(r)}\right)^{\top}\left(\widehat{\mathbf{H}}_{\widehat{\mathbf{A}}}^{(r)}\mathbf{W}^{(t-1)} - \mathbf{Y}\right). \tag{4}
$$

It follows by (4) that

$$
\widehat{\mathbf{H}}_{\widehat{\mathbf{A}}}^{(r)}\mathbf{W}^{(t)} = \mathbf{H}_{\widehat{\mathbf{A}}}^{(r)}\mathbf{W}^{(t-1)} - \eta\mathbf{K}^{(r)}\left(\widehat{\mathbf{A}}\mathbf{H}\mathbf{W}^{(t-1)} - \mathbf{Y}\right), \tag{5}
$$

where $\mathbf{K}^{(r)} = \mathbf{H}_{\widehat{\mathbf{A}}}^{(r)}\left(\mathbf{H}_{\widehat{\mathbf{A}}}^{(r)}\right)^{\top}$, and recall that $\mathbf{K}^{(r_0)} = \mathbf{K} = \mathbf{H}_{\widehat{\mathbf{A}}}\mathbf{H}_{\widehat{\mathbf{A}}}^{\top}$ where $r_0$ is the rank of the feature matrix $\mathbf{H}_{\widehat{\mathbf{A}}}$.

With $\mathbf{F}(\mathbf{W}, r, t) = \mathbf{H}_{\widehat{\mathbf{A}}}^{(r)}\mathbf{W}^{(t)}$, it follows by (5) and the fact that $\mathbf{Y}^{\perp}(r)$ is orthogonal to the to top-$r$ eigenvectors of $\mathbf{K}^{(r)}$ that

$$
\mathbf{F}(\mathbf{W}, r, t) - \bar{\mathbf{Y}}(r) = \left(\mathbf{I}_N - \eta\mathbf{K}^{(r)}\right)\left(\mathbf{F}(\mathbf{W}, r, t) - \bar{\mathbf{Y}}(r)\right),
$$

where $\bar{\mathbf{Y}}(r) = \mathbf{Y} - \mathbf{Y}^{\perp}(r)$. It follows that

$$
\mathbf{F}(\mathbf{W}, r, t) - \bar{\mathbf{Y}}(r) = \left(\mathbf{I}_N - \eta\mathbf{K}^{(r)}\right)^t\left(\mathbf{F}(\mathbf{W}, r, 0) - \bar{\mathbf{Y}}(r)\right) = -\left(\mathbf{I}_N - \eta\mathbf{K}^{(r)}\right)^t\bar{\mathbf{Y}}(r),
$$

and

$$
\left\|\mathbf{F}(\mathbf{W}, r, t) - \bar{\mathbf{Y}}(r)\right\|_{\mathrm{F}} \leq \left(1 - \eta\widehat{\lambda}_r^2\right)^t\left\|\bar{\mathbf{Y}}(r)\right\|_2 \leq \left(1 - \eta\widehat{\lambda}_r^2\right)^t\|\mathbf{Y}\|_{\mathrm{F}} \tag{6}
$$

As a result of (7), we have

$$
\left\|\mathbf{F}(\mathbf{W}, r, t) - \bar{\mathbf{Y}}(r)_{\mathcal{L}}\right\|_{\mathrm{F}} \leq \left(1 - \eta\widehat{\lambda}_r^2\right)^t\left\|\bar{\mathbf{Y}}(r)\right\|_{\mathrm{F}} \leq \left(1 - \eta\widehat{\lambda}_r^2\right)^t\|\mathbf{Y}\|_{\mathrm{F}} \tag{7}
$$

Using $\bar{\mathbf{Y}}(r)$ as the label, we apply Theorem 3.8 in (Yang) that

$$
\frac{1}{u}\left\|\left[\mathbf{F}(\mathbf{W}, r, t) - \bar{\mathbf{Y}}(r)\right]_{\mathcal{U}}\right\|_{\mathrm{F}}^2 \leq \left(1 + \frac{1}{c}\right)\frac{1}{m}\left\|\left[\mathbf{F}(\mathbf{W}, r, t) - \bar{\mathbf{Y}}(r)\right]_{\mathcal{L}}\right\|_{\mathrm{F}}^2 + c_1 c_3 r\left(\frac{1}{u} + \frac{1}{m}\right) + \frac{c_2 x}{u}, \tag{8}
$$

where $c_1, c_2, c_3$ are positive numbers depending on $\mathbf{U}$, $\left\{\widehat{\lambda}_i\right\}_{i=1}^r$, and $\tau_0$ with $\tau_0^2 = \max_{i \in [N]}\mathbf{K}_{ii}$.

It follows by (7) and (8) that

$$
\begin{aligned}
&\frac{1}{u}\left\|\left[\mathbf{F}(\mathbf{W}, r, t) - \bar{\mathbf{Y}}(r)\right]_{\mathcal{U}}\right\|_{\mathrm{F}}^2 \\
&\leq \frac{1 + 1/c}{m}\left\|\left[\mathbf{F}(\mathbf{W}, r, t) - \bar{\mathbf{Y}}(r)\right]_{\mathcal{L}}\right\|_2^2 + c_1 c_3 r\left(\frac{1}{u} + \frac{1}{m}\right) + \frac{c_2 x}{u} + \left\|\mathbf{Y}^{\perp}(r)\right\|_{\mathrm{F}}^2 \\
&\leq \frac{1 + 1/c}{m}\left(1 - \eta\widehat{\lambda}_r^2\right)^{2t}\|\mathbf{Y}\|_{\mathrm{F}}^2 + c_1 c_3 r\left(\frac{1}{u} + \frac{1}{m}\right) + \frac{c_2 x}{u},
\end{aligned} \tag{9}
$$

which completes the proof.

$\square$

## B  DETAILS ABOUT DPMM

We propose Bayesian nonparametric Prototype Learning (BPL) to infer the pseudo labels, or cluster labels, of nodes. BPL, as a Bayesian nonparametric algorithm, infers the cluster prototypes by the Dirichlet Process Mixture Model (DPMM) under the assumption that the distribution of node representations is a mixture of Gaussians. The Gaussians share the same fixed covariance matrix $\sigma\mathbf{I}$, and each Gaussian is used to model a cluster. The DPMM model is specified by $G \sim \text{DP}(G_0, \alpha), \phi_i \sim G, \mathbf{H}_i \sim \mathcal{N}(\phi_i, \sigma\mathbf{I}), \; i = 1, ..., N$, where $G$ is a Gaussian distribution draw from the Dirichlet process $\text{DP}(G_0, \alpha)$, and $\alpha$ is the concentration parameter for $\text{DP}(G_0, \alpha)$. $\phi_i$ is the mean of the Gaussian sampled for generating the node representation $\mathbf{H}_i$. $G_0$ is the prior over means of the Gaussians. $G_0$ is set to a zero-mean Gaussian $\mathcal{N}(\mathbf{0}, \rho\mathbf{I})$ for $\rho > 0$. A collapsed Gibbs sampler (Resnik & Hardisty, 2010) is used to infer the components of the GMM with the DPMM. The Gibbs sampler iteratively samples pseudo labels for the nodes given the means of the Gaussian components, and samples the means of the Gaussian components given the pseudo labels of the nodes. Let $\tilde{K}$ denote the number of inferred prototypes at the current iteration, the pseudo label $z_i$ of node $v_i$ is then calculated by $z_i = \arg\min_k \{d_{ik}\}$, where $d_{ik} = \|\mathbf{H}_i - \mathbf{c}_k\|_2^2$ for $k = 1, ..., \tilde{K}$, and $d_{ik} = \xi$ for $k = \tilde{K} + 1$. $\xi$ is the margin to initialize a new prototype. In practice, we choose the value of $\xi$ by performing cross-validation on each dataset.

## C  IMPLEMENTATION DETAILS

### C.1  DATASETS

We evaluate BRGCL on eight public benchmarks that are widely used for node representation learning, namely Cora, Citeseer, PubMed (Sen et al., 2008), Coauthor CS, ogbn-arxiv (Hu et al., 2020), Wiki-CS (Mernyei & Cangea, 2020), Amazon-Computers, and Amazon-Photos (Shchur et al., 2018). Cora, Citeseer, and PubMed are the three most widely used citation networks. Coauthor CS is a co-authorship graph. The ogbn-arxiv is a directed citation graph. Wiki-CS is a hyperlink networks of computer science articles. Amazon-Computers and Amazon-Photos are co-purchase networks of products selling on Amazon.com. We summarize the statistics of all the datasets in Table 2. For all our experiments, we follow the default separation (Shchur et al., 2018; Mernyei & Cangea, 2020; Hu et al., 2020) of training, validation, and test sets on each benchmark.

Table 2: The statistics of the datasets.

| Dataset | Nodes | Edges | Features | Classes |
|---|---|---|---|---|
| Cora | 2,708 | 5,429 | 1,433 | 7 |
| CiteSeer | 3,327 | 4,732 | 3,703 | 6 |
| PubMed | 19,717 | 44,338 | 500 | 3 |
| Coauthor CS | 18,333 | 81,894 | 6,805 | 15 |
| ogbn-arxiv | 169,343 | 1,166,243 | 128 | 40 |
| Wiki-CS | 11,701 | 215,863 | 300 | 10 |
| Amazon-Computers | 13,752 | 245,861 | 767 | 10 |
| Amazon-Photos | 7,650 | 119,081 | 745 | 8 |

### C.2  COMPARED METHODS

To demonstrate the power of LR-RGCL in learning robust node representation, we compare LR-RGCL with two robust contrastive learning baselines, Jo-SRC and Sel-CL, which select clean samples for image data. Since their sample selection methods are general and not limited to the image domain, we adopt these two baselines in our experiments. Next, we introduce the implementation details of applying Jo-SRC and Sel-CL on graph data.

**Jo-SRC** (Yao et al., 2021): Jo-SRC is a robust contrastive learning method proposed for image classification. It selects clean samples for training by adopting the Jensen-Shannon divergence to measure the likelihood of each sample being clean. Because this method is a general selection strategy on the representation space, it is adapted to selecting clean samples in the representation space of nodes in GCL. It also introduces a consistency regularization term to the contrastive loss to

improve the robustness. To get a competitive robust GCL baseline, we apply the sample selection strategy and the consistency regularization proposed by Jo-SRC to state-of-the-art GCL methods MVGRL, MERIT, and SUGRL. We add the regularization term in Jo-SRC to the graph contrastive loss. The GCL encoders are trained only on the clean samples selected by Jo-SRC. We only report the best results for comparison, which are achieved by applying Jo-SRC to MERIT.

**Sel-CL** (Li et al., 2022): Sel-CL is a supervised contrastive learning proposed to learn robust pre-trained representations for image classification. It proposes to select confident contrastive pairs in the contrastive learning frameworks. Sel-CL first selects confident examples by measuring the agreement between learned representations and labels generated by label propagation with cross-entropy loss. Next, Sel-CL selects contrastive pairs from those with selected confident examples in them. This method is also a general sample selection strategy on a learned representation space. So we can adapt Sel-CL to the node representation space to select confident pairs for GCL. In this process, they only select contrastive pairs whose representation similarity is higher than a dynamic threshold. In our experiments, we also adopt the confident contrastive pair selection strategy to the state-of-the-art GCL methods MVGRL, MERIT, and SUGRL. With the same GCL framework, GCL encoders are trained only on the confident pairs selected by Sel-CL. We only report the best results for comparison, which are achieved by applying Sel-CL to MERIT.

## D    ADDITIONAL EXPERIMENTAL RESULTS

### D.1    TUNING HYPER-PARAMETERS BY CROSS-VALIDATION

In this section, we show the tuning procedures on the hyper-parameters $\xi$ and $\gamma_0$ in Table 3. We perform cross-validations on $20\%$ of training data to decide the value of $\xi$ and $\gamma_0$. The value of $\xi$ is selected from $\{0.1, 0.15, 0.2, 0.25, 0.3, 0.35, 0.4, 0.45, 0, 5\}$. The value of $\gamma_0$ is selected from $\{0.1, 0.2, 0.3, 0.4, 0.5, 0.6, 0.7, 0.8, 0.9\}$. The selected values for $\xi$ and $\gamma_0$ on each dataset are shown in Table 3.

Table 3: Selected hyper-parameters for each dataset.

| Dataset | Cora | Citeseer | PubMed | Coauthor CS | ogbn-arxiv | Wiki-CS | Amazon-Computers | Amazon-Photos |
|---------|------|----------|--------|-------------|------------|---------|------------------|---------------|
| $\xi$ | 0.20 | 0.15 | 0.35 | 0.40 | 0.25 | 0.35 | 0.25 | 0.25 |
| $\gamma_0$ | 0.3 | 0.5 | 0.7 | 0.4 | 0.4 | 0.7 | 0.5 | 0.5 |

### D.2    NODE CLASSIFICATION ON WIKI-CS, AMAZON-COMPUTERS, AND AMAZON-PHOTOS

The results for node classification with symmetric label noise, asymmetric label noise, and attribute noise on Wiki-CS, Amazon-Computers, and Amazon-Photos are shown in Table 4. It is observed that LR-RGCL also outperforms all the baselines for node classification with both label noise and attribute noise on these three benchmark datasets.

### D.3    NODE CLASSIFICATION RESULTS FOR GCL METHODS WITH DIFFERENT TYPES OF CLASSIFIERS

Existing GCL methods, such as MERIT, SUGRL, and SFA, first train a graph encoder with graph contrastive learning objectives such as InfoNCE (Jin et al., 2021). After obtaining the node representation learned by contrastive learning, a linear layer for classification is trained in the supervised setting. In contrast, LR-RGCL adopts a transductive classifier on top of the node representation obtained by contrastive learning. For fair comparisons with previous GCL methods, we also train the compared GCL baselines with the same transductive classifier as in LR-RGCL and a two-layer transductive GCN classifier. The results with different types of classifiers are deferred in Section D.3 of the supplementary.

### D.4    TRAINING TIME COMPARISONS AND STUDY ON DIFFERENT RANKS

In this section, we first compare the training time of LR-RGCL against other baseline methods on all benchmark datasets. For our method, we include the training time of robust graph contrastive

Table 4: Performance comparison for node classification on Wiki-CS, Amazon-Computers, and Amazon-Photos with asymmetric label noise, symmetric label noise, and attribute noise.

| Dataset | Methods | Noise Level 0 | 40 Asymmetric | 40 Symmetric | 40 Attribute | 60 Asymmetric | 60 Symmetric | 60 Attribute | 80 Asymmetric | 80 Symmetric | 80 Attribute |
|---|---|---|---|---|---|---|---|---|---|---|---|
| Wiki-CS | GCN | 0.918±0.001 | 0.645±0.009 | 0.656±0.006 | 0.702±0.010 | 0.511±0.013 | 0.501±0.009 | 0.531±0.010 | 0.429±0.022 | 0.389±0.011 | 0.415±0.013 |
| | S²GC | 0.918±0.001 | 0.657±0.012 | 0.663±0.006 | 0.713±0.010 | 0.516±0.013 | 0.514±0.009 | 0.556±0.009 | 0.437±0.020 | 0.396±0.010 | 0.422±0.012 |
| | GCE | 0.922±0.003 | 0.662±0.017 | 0.659±0.007 | 0.705±0.014 | 0.515±0.016 | 0.502±0.007 | 0.539±0.009 | 0.443±0.017 | 0.389±0.012 | 0.412±0.011 |
| | UnionNET | 0.918±0.002 | 0.669±0.023 | 0.671±0.013 | 0.706±0.012 | 0.525±0.011 | 0.529±0.011 | 0.540±0.012 | 0.458±0.015 | 0.401±0.011 | 0.420±0.007 |
| | NRGNN | 0.919±0.002 | 0.678±0.014 | 0.689±0.009 | 0.705±0.012 | 0.545±0.021 | 0.556±0.011 | 0.546±0.011 | 0.461±0.012 | 0.410±0.012 | 0.417±0.007 |
| | RTGNN | 0.920±0.005 | 0.678±0.012 | 0.691±0.009 | 0.712±0.008 | 0.559±0.010 | 0.569±0.011 | 0.560±0.008 | 0.455±0.015 | 0.415±0.015 | 0.412±0.014 |
| | SUGRL | 0.922±0.005 | 0.675±0.010 | 0.695±0.010 | 0.714±0.006 | 0.550±0.011 | 0.560±0.011 | 0.561±0.007 | 0.449±0.011 | 0.411±0.011 | 0.429±0.008 |
| | MERIT | 0.924±0.004 | 0.679±0.011 | 0.689±0.008 | 0.709±0.005 | 0.552±0.014 | 0.562±0.014 | 0.562±0.011 | 0.452±0.013 | 0.403±0.013 | 0.426±0.005 |
| | ARIEL | 0.925±0.004 | 0.682±0.011 | 0.699±0.009 | 0.712±0.005 | 0.555±0.011 | 0.566±0.011 | 0.556±0.011 | 0.454±0.014 | 0.415±0.019 | 0.427±0.013 |
| | SFA | 0.925±0.009 | 0.682±0.011 | 0.690±0.012 | 0.715±0.012 | 0.555±0.015 | 0.567±0.014 | 0.565±0.013 | 0.458±0.013 | 0.402±0.013 | 0.429±0.015 |
| | Sel-Cl | 0.922±0.008 | 0.684±0.009 | 0.694±0.012 | 0.714±0.010 | 0.557±0.013 | 0.568±0.013 | 0.566±0.010 | 0.457±0.013 | 0.412±0.017 | 0.425±0.009 |
| | Jo-SRC | 0.921±0.005 | 0.684±0.011 | 0.695±0.010 | 0.709±0.007 | 0.560±0.011 | 0.566±0.011 | 0.561±0.009 | 0.456±0.013 | 0.410±0.018 | 0.428±0.010 |
| | GRAND+ | 0.927±0.004 | 0.682±0.011 | 0.693±0.006 | 0.715±0.008 | 0.554±0.008 | 0.568±0.013 | 0.557±0.011 | 0.455±0.012 | 0.416±0.013 | 0.428±0.011 |
| | RGCL | 0.929±0.006 | 0.694±0.013 | 0.718±0.008 | 0.733±0.009 | 0.570±0.014 | 0.587±0.011 | 0.585±0.012 | 0.465±0.012 | 0.434±0.015 | 0.444±0.012 |
| | LR-RGCL | **0.933±0.006** | **0.699±0.015** | **0.721±0.011** | **0.742±0.015** | **0.575±0.014** | **0.595±0.018** | **0.588±0.015** | **0.469±0.015** | **0.438±0.015** | **0.453±0.017** |
| Amazon-Computers | GCN | 0.815±0.005 | 0.547±0.015 | 0.636±0.007 | 0.639±0.008 | 0.405±0.014 | 0.517±0.010 | 0.439±0.012 | 0.265±0.012 | 0.354±0.014 | 0.317±0.013 |
| | S²GC | 0.835±0.002 | 0.569±0.014 | 0.664±0.007 | 0.661±0.007 | 0.422±0.010 | 0.535±0.010 | 0.454±0.011 | 0.279±0.014 | 0.366±0.014 | 0.320±0.013 |
| | GCE | 0.819±0.004 | 0.573±0.011 | 0.652±0.008 | 0.650±0.014 | 0.449±0.011 | 0.509±0.011 | 0.445±0.015 | 0.280±0.013 | 0.353±0.013 | 0.325±0.015 |
| | UnionNET | 0.820±0.006 | 0.569±0.014 | 0.664±0.007 | 0.653±0.012 | 0.452±0.010 | 0.541±0.010 | 0.450±0.009 | 0.283±0.014 | 0.370±0.011 | 0.320±0.012 |
| | NRGNN | 0.822±0.006 | 0.571±0.019 | 0.676±0.007 | 0.645±0.012 | 0.470±0.014 | 0.548±0.014 | 0.451±0.011 | 0.282±0.022 | 0.373±0.012 | 0.326±0.010 |
| | RTGNN | 0.828±0.003 | 0.570±0.010 | 0.682±0.008 | 0.678±0.011 | 0.474±0.011 | 0.555±0.010 | 0.457±0.009 | 0.280±0.011 | 0.386±0.014 | 0.342±0.016 |
| | SUGRL | 0.834±0.005 | 0.564±0.011 | 0.674±0.012 | 0.675±0.009 | 0.468±0.011 | 0.552±0.011 | 0.452±0.012 | 0.280±0.012 | 0.381±0.012 | 0.338±0.014 |
| | MERIT | 0.831±0.005 | 0.560±0.008 | 0.670±0.008 | 0.671±0.009 | 0.467±0.013 | 0.547±0.013 | 0.450±0.014 | 0.277±0.013 | 0.385±0.013 | 0.335±0.009 |
| | ARIEL | 0.843±0.004 | 0.573±0.013 | 0.681±0.010 | 0.675±0.009 | 0.471±0.012 | 0.553±0.012 | 0.455±0.014 | 0.284±0.014 | 0.389±0.013 | 0.343±0.013 |
| | SFA | 0.839±0.010 | 0.564±0.011 | 0.677±0.013 | 0.676±0.015 | 0.473±0.014 | 0.549±0.014 | 0.457±0.014 | 0.282±0.016 | 0.389±0.013 | 0.344±0.017 |
| | Sel-Cl | 0.828±0.002 | 0.570±0.010 | 0.685±0.012 | 0.676±0.009 | 0.472±0.013 | 0.554±0.014 | 0.455±0.011 | 0.282±0.017 | 0.389±0.013 | 0.341±0.015 |
| | Jo-SRC | 0.825±0.005 | 0.571±0.006 | 0.684±0.013 | 0.679±0.007 | 0.473±0.011 | 0.556±0.008 | 0.458±0.012 | 0.285±0.013 | 0.387±0.018 | 0.345±0.018 |
| | GRAND+ | 0.858±0.006 | 0.570±0.009 | 0.682±0.007 | 0.678±0.011 | 0.472±0.010 | 0.554±0.008 | 0.456±0.012 | 0.284±0.015 | 0.387±0.015 | 0.345±0.013 |
| | RGCL | 0.854±0.006 | 0.584±0.009 | 0.704±0.007 | 0.690±0.010 | 0.484±0.013 | 0.577±0.013 | 0.469±0.013 | 0.295±0.012 | 0.407±0.012 | 0.356±0.011 |
| | LR-RGCL | **0.858±0.006** | **0.589±0.011** | **0.713±0.007** | **0.695±0.011** | **0.492±0.011** | **0.587±0.013** | **0.477±0.012** | **0.306±0.012** | **0.419±0.012** | **0.363±0.011** |
| Amazon-Photos | GCN | 0.703±0.005 | 0.475±0.023 | 0.501±0.013 | 0.529±0.009 | 0.351±0.014 | 0.341±0.014 | 0.372±0.011 | 0.291±0.022 | 0.281±0.019 | 0.290±0.014 |
| | S²GC | 0.736±0.005 | 0.488±0.013 | 0.528±0.013 | 0.553±0.008 | 0.363±0.012 | 0.367±0.014 | 0.390±0.013 | 0.304±0.024 | 0.284±0.019 | 0.288±0.011 |
| | GCE | 0.705±0.004 | 0.490±0.016 | 0.512±0.014 | 0.540±0.014 | 0.362±0.015 | 0.352±0.010 | 0.381±0.009 | 0.309±0.012 | 0.285±0.014 | 0.285±0.011 |
| | UnionNET | 0.706±0.006 | 0.499±0.015 | 0.547±0.014 | 0.545±0.013 | 0.379±0.013 | 0.399±0.013 | 0.379±0.012 | 0.322±0.021 | 0.302±0.013 | 0.290±0.012 |
| | NRGNN | 0.710±0.006 | 0.498±0.015 | 0.546±0.015 | 0.538±0.011 | 0.382±0.016 | 0.412±0.016 | 0.377±0.012 | 0.336±0.021 | 0.309±0.018 | 0.284±0.009 |
| | RTGNN | 0.746±0.008 | 0.498±0.007 | 0.556±0.007 | 0.550±0.012 | 0.392±0.010 | 0.424±0.013 | 0.390±0.014 | 0.348±0.017 | 0.308±0.016 | 0.302±0.011 |
| | SUGRL | 0.730±0.005 | 0.493±0.011 | 0.541±0.011 | 0.544±0.010 | 0.376±0.009 | 0.421±0.009 | 0.388±0.009 | 0.339±0.010 | 0.305±0.010 | 0.300±0.009 |
| | MERIT | 0.740±0.007 | 0.496±0.012 | 0.536±0.012 | 0.542±0.010 | 0.383±0.011 | 0.425±0.011 | 0.387±0.008 | 0.344±0.014 | 0.301±0.014 | 0.295±0.009 |
| | SFA | 0.740±0.011 | 0.502±0.014 | 0.532±0.015 | 0.547±0.013 | 0.390±0.014 | 0.433±0.014 | 0.389±0.012 | 0.347±0.016 | 0.312±0.015 | 0.299±0.013 |
| | ARIEL | 0.727±0.007 | 0.500±0.008 | 0.550±0.013 | 0.548±0.008 | 0.391±0.009 | 0.427±0.012 | 0.389±0.014 | 0.349±0.014 | 0.307±0.013 | 0.299±0.013 |
| | Sel-Cl | 0.725±0.008 | 0.499±0.012 | 0.551±0.010 | 0.549±0.008 | 0.389±0.011 | 0.426±0.008 | 0.391±0.020 | 0.350±0.018 | 0.310±0.015 | 0.300±0.017 |
| | Jo-SRC | 0.730±0.005 | 0.500±0.013 | 0.555±0.011 | 0.551±0.011 | 0.394±0.013 | 0.425±0.013 | 0.393±0.013 | 0.351±0.013 | 0.305±0.018 | 0.303±0.013 |
| | GRAND+ | 0.756±0.004 | 0.497±0.010 | 0.553±0.010 | 0.552±0.011 | 0.390±0.013 | 0.422±0.013 | 0.387±0.013 | 0.348±0.013 | 0.309±0.014 | 0.302±0.012 |
| | RGCL | 0.748±0.009 | 0.510±0.013 | 0.574±0.013 | 0.562±0.007 | 0.403±0.014 | 0.445±0.014 | 0.399±0.012 | 0.359±0.012 | 0.327±0.014 | 0.312±0.010 |
| | LR-RGCL | **0.757±0.010** | **0.520±0.013** | **0.581±0.013** | **0.570±0.007** | **0.410±0.014** | **0.455±0.014** | **0.406±0.012** | **0.369±0.012** | **0.335±0.014** | **0.318±0.010** |

Table 5: Performance comparison for node classification by inductive linear classifier, transductive two-layer GCN classifier, and transductive classifier used in LR-RGCL. The comparisons are performed on Cora with asymmetric label noise, symmetric label noise, and attribute noise.

| Methods | Noise Level 0 | 40 Asymmetric | 40 Symmetric | 40 Attribute | 60 Asymmetric | 60 Symmetric | 60 Attribute | 80 Asymmetric | 80 Symmetric | 80 Attribute |
|---|---|---|---|---|---|---|---|---|---|---|
| SUGRL (original, inductive classifier) | 0.834±0.005 | 0.564±0.011 | 0.674±0.012 | 0.675±0.009 | 0.468±0.011 | 0.552±0.011 | 0.452±0.012 | 0.280±0.012 | 0.381±0.012 | 0.338±0.014 |
| SUGRL + transductive GCN | 0.833±0.006 | 0.562±0.013 | 0.675±0.015 | 0.673±0.012 | 0.470±0.011 | 0.551±0.011 | 0.454±0.012 | 0.280±0.012 | 0.380±0.012 | 0.340±0.014 |
| SUGRL + linear transductive classifier | 0.836±0.007 | 0.568±0.013 | 0.677±0.010 | 0.674±0.011 | 0.472±0.011 | 0.555±0.011 | 0.457±0.012 | 0.284±0.012 | 0.383±0.012 | 0.341±0.014 |
| MERIT (original, inductive classifier) | 0.831±0.005 | 0.560±0.008 | 0.670±0.008 | 0.671±0.009 | 0.467±0.013 | 0.547±0.013 | 0.450±0.014 | 0.277±0.013 | 0.385±0.013 | 0.335±0.009 |
| MERIT + transductive GCN | 0.831±0.007 | 0.562±0.011 | 0.668±0.013 | 0.672±0.014 | 0.466±0.013 | 0.549±0.015 | 0.451±0.016 | 0.276±0.012 | 0.382±0.014 | 0.337±0.013 |
| MERIT + linear transductive classifier | 0.833±0.003 | 0.562±0.014 | 0.673±0.012 | 0.673±0.011 | 0.466±0.015 | 0.546±0.016 | 0.453±0.017 | 0.280±0.016 | 0.386±0.011 | 0.336±0.014 |
| SFA (original, inductive classifier) | 0.839±0.010 | 0.564±0.011 | 0.677±0.013 | 0.676±0.015 | 0.473±0.014 | 0.549±0.014 | 0.457±0.014 | 0.282±0.016 | 0.389±0.013 | 0.344±0.017 |
| SFA + transductive GCN | 0.837±0.013 | 0.565±0.011 | 0.673±0.017 | 0.673±0.018 | 0.474±0.016 | 0.551±0.015 | 0.453±0.018 | 0.277±0.016 | 0.389±0.015 | 0.343±0.019 |
| SFA + linear transductive classifier | 0.841±0.015 | 0.566±0.013 | 0.678±0.014 | 0.679±0.014 | 0.477±0.015 | 0.552±0.012 | 0.456±0.016 | 0.284±0.017 | 0.391±0.015 | 0.348±0.019 |
| LR-RGCL | **0.858±0.006** | **0.589±0.011** | **0.713±0.007** | **0.695±0.011** | **0.492±0.011** | **0.587±0.013** | **0.477±0.012** | **0.306±0.012** | **0.419±0.012** | **0.363±0.011** |

learning, the time of the SVD computation of the kernel, and the training time of the transductive classifier. For graph contrastive learning methods, we include both the training time of the GCL encoder and the downstream classifier. We evaluate the training time on one 80 GB A100 GPU. The results are shown in Table 7. It is observed that the LR-RGCL takes similar training time as baseline GCL methods such as SFA and MERIT.

We also perform ablation study on the value of rank $r$ in the optimization problem (2) for our low-rank transductive classifier. It is observed from Table 6 that the performance of our low-rank classifier is consistently close to the best performance among all the choices of the rank when $r$ is between $0.1 \min \{N, d\}$ and $0.2 \min \{N, d\}$.

## D.5 EIGEN-PROJECTION AND CONCENTRATION ENTROPY

Figure 3 illustrates the eigen-projection and energy concentration ratio for more datasets.

## D.6 VISUALIZATION OF CONFIDENCE SCORE

We visualize the confident nodes selected by BPL in the embedding space of the learned node representations in Figure 4. The node representations are visualized by the t-SNE figure. Each mark

Table 6: Ablation study on the value of rank $r$ in the optimization problem (2) on Cora with different levels of asymmetric and symmetric label noise. The accuracy with the optimal rank is shown in the last row. The accuracy difference against the optimal rank is shown for other ranks.

| | Noise Level | | | | | | |
|---|---|---|---|---|---|---|---|
| Rank | 0 | 40 | | 60 | | 80 | |
| | - | Asymmetric | Symmetric | Asymmetric | Symmetric | Asymmetric | Symmetric |
| $0.1 \min\{N, d\}$ | -0.002 | -0.001 | -0.002 | -0.002 | -0.001 | -0.001 | -0.000 |
| $0.2 \min\{N, d\}$ | -0.000 | -0.000 | -0.000 | -0.000 | -0.000 | -0.000 | -0.000 |
| $0.3 \min\{N, d\}$ | -0.000 | -0.000 | -0.001 | -0.002 | -0.001 | -0.000 | -0.001 |
| $0.4 \min\{N, d\}$ | -0.001 | -0.003 | -0.002 | -0.001 | -0.002 | -0.002 | -0.002 |
| $0.5 \min\{N, d\}$ | -0.001 | -0.002 | -0.003 | -0.003 | -0.003 | -0.001 | -0.002 |
| $0.6 \min\{N, d\}$ | -0.003 | -0.002 | -0.002 | -0.003 | -0.002 | -0.002 | -0.003 |
| $0.7 \min\{N, d\}$ | -0.003 | -0.004 | -0.003 | -0.004 | -0.004 | -0.004 | -0.005 |
| $0.8 \min\{N, d\}$ | -0.002 | -0.005 | -0.006 | -0.006 | -0.006 | -0.007 | -0.007 |
| $0.9 \min\{N, d\}$ | -0.004 | -0.004 | -0.005 | -0.007 | -0.008 | -0.008 | -0.006 |
| $\min\{N, d\}$ | -0.004 | -0.004 | -0.007 | -0.007 | -0.008 | -0.010 | -0.008 |
| optimal | 0.858 | 0.589 | 0.713 | 0.492 | 0.587 | 0.306 | 0.419 |

| Methods | Cora | Citeseer | PubMed | Coauthor-CS | Wiki-CS | Computer | Photo | ogbn-arxiv |
|---|---|---|---|---|---|---|---|---|
| GCN | 11.5 | 13.7 | 38.6 | 43.2 | 22.3 | 30.2 | 19.0 | 215.1 |
| S$^2$GC | 20.7 | 22.5 | 47.2 | 57.2 | 27.6 | 38.5 | 22.2 | 243.7 |
| GCE | 32.6 | 36.9 | 67.3 | 80.8 | 37.6 | 50.1 | 32.2 | 346.1 |
| UnionNET | 67.5 | 69.7 | 100.5 | 124.2 | 53.2 | 69.2 | 45.3 | 479.3 |
| NRGNN | 72.4 | 80.5 | 142.7 | 189.4 | 74.3 | 97.2 | 62.4 | 650.2 |
| RTGNN | 143.3 | 169.5 | 299.5 | 353.5 | 153.7 | 201.5 | 124.2 | 1322.2 |
| SUGRL | 100.3 | 122.1 | 207.4 | 227.1 | 107.7 | 142.8 | 87.7 | 946.8 |
| MERIT | 167.2 | 179.2 | 336.7 | 375.3 | 172.3 | 226.5 | 140.6 | 1495.1 |
| ARIEL | 156.9 | 164.3 | 284.3 | 332.6 | 145.1 | 190.4 | 118.3 | 1261.4 |
| SFA | 237.5 | 269.4 | 457.1 | 492.3 | 233.5 | 304.5 | 187.2 | 2013.1 |
| Sel-Cl | 177.3 | 189.9 | 313.5 | 352.5 | 161.7 | 211.1 | 130.9 | 1401.1 |
| Jo-SRC | 148.2 | 157.1 | 281.0 | 306.1 | 144.5 | 188.0 | 118.5 | 1256.0 |
| GRAND+ | 57.4 | 68.4 | 101.7 | 124.2 | 54.8 | 73.8 | 44.5 | 479.2 |
| LR-RGCL | 159.9 | 174.5 | 350.7 | 380.9 | 180.3 | 235.7 | 145.5 | 1552.7 |

Table 7: Training time (seconds) comparisons for node classification.

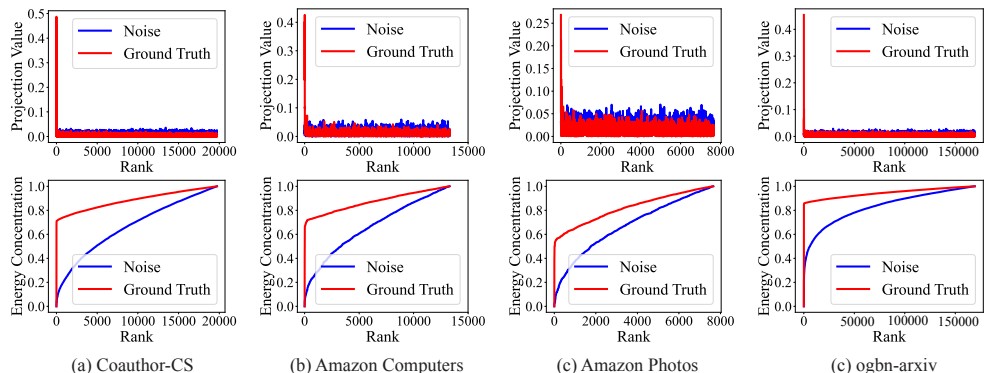

Figure 3: Eigen-projection (first row) and energy concentration (second row) on Coauthor-CS, Amazon-Computers, Amazon-Photos, and ogbn-arxiv. By the rank of $0.2n$, the concentration entropy on Coauthor-CS, Amazon-Computers, Amazon-Photos, and ogbn-arxiv are $0.779$, $0.809$, $0.752$, and $0.787$.

in t-SNE represents the representation of a node, and the color of the mark denotes the confidence of that node. The results are shown for different levels of attribute noise. It is observed from Figure 4 that confident nodes, which are redder in Figure 4, are well separated in the embedding space. With a higher level of attribute noise, the bluer nodes from different clusters blended around the cluster boundaries. In contrast, the redder nodes are still well separated and far away from cluster boundaries, which leads to more robustness and better performance in downstream tasks.

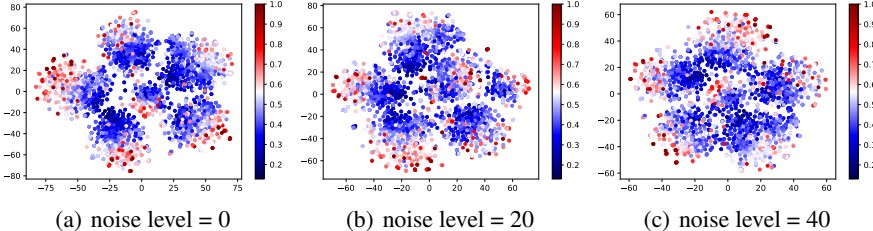

Figure 4: Visualization of confident nodes with different levels of attribute noise for semi-supervised node classification on Citeseer.

