# OpenReview forum: "Low-Rank Robust Graph Contrastive Learning"
_ICLR.cc/2024/Conference — ICLR 2024 Conference Withdrawn Submission_

### Official Review · Reviewer_wZD8 · 2023-10-23

**Soundness:** 2 fair
**Presentation:** 1 poor
**Contribution:** 2 fair
**Rating:** 3
**Confidence:** 4

**Summary:**

This paper proposes a robust graph contrastive learning framework based on prototype learning and low-rank decomposition. The key ideas are to explore the high-confident prototypical nodes for contrastive learning and utilize the low-rank structure of the kernel matrix for denoising. Bayesian nonparametric prototype learning is adopted to identify high-confident nodes. SVD decomposition is performed on the kernel matrix generated from the RGCL embeddings, based on which a linear classifier is trained. Experiments on real world datasets validate the effectiveness and robustness of the proposed method.

**Strengths:**

-	The idea of utilizing BPL and low-rank for robust graph ML is interesting.
-	Theoretical analysis on the generalization bound is provided to assess the effectiveness of the proposed method.
-	Experiments on real-world graphs seem solid. Source code and hyperparameter settings are provided for reproducibility check.

**Weaknesses:**

-	More ablation studies needed. Since the model contains two modules (BPL based prototype learning and low-rank classifier), which part contributes more to the effectiveness and robustness?
-	Presentation. Overall, I think further improvements on presentation are needed. The current writing spent too much space for less important preliminaries and related works, while too little space for the method (e.g., for DPMM, label propagation, BPL, etc.). See more details in the Question part.

**Questions:**

-	For Eq. (2), can you explain why KL divergence is adopted here? Since $y_i$ is a one-hot vector, the calculation for KL divergence will be 0 for all the zero elements in $y_i$.
-	Current framework utilizes the BPL and prototype CL separately. Since both BPL and low-rank aims to deal with the noise/robustness, will it be possible, and potentially beneficial, to incorporate both BPL and low-rank property simultaneously?
-	Theorem 4.1 provide a upper bound on the Frobenius distance between F(W,r,t) and $\overline{Y}$. It would be beneficial to derive a distance between F(w,r,t) and $\tilde{Y}$ to measure the difference between prediction and the noise-free label.
-	Based on the definition of $\mathcal{T}_k$ at the end of page 5, is there a lot of overlap between different $\mathcal{T}_k, \forall k=1,…,K$? According to my understanding, if $\gamma_i>\gamma_j$, then $\mathcal{T}_j\subset\mathcal{T}_i$?
-	From Figure 4, it seems that high-confident prototype nodes are actually quite far away from the cluster center. My concern is whether they can well represent the cluster? If not, the objective function $L_{proto}$ encouraging node embeddings to be similar to the prototype nodes may push the clusters to the underrepresented prototype nodes.
-	Presentation:
  - Eq.(1): What is c_k in the nominator? Consider use another subscript to distinguish c_k in the numerator from c_k in the denominator.
  - For $c_k$ defined at the end of Section 4.2, For nodes in $\mathcal{T}_k$, they are not necessarily connected (and even be distant), I’m wondering whether such mean aggregation is valid? E.g., what if the node features are positional embeddings? Such mean aggregation of distant and disconnected nodes will corrupt the information.
  - Shrink the caption for Fig.1, and move some technical parts, e.g., how to compute p, etc, to the main paragraphs.
  - Font sizes are too small for Table 1,4,5.
  - Typo in abstract: Robst -> Robust; missing superscript $(r)$ for the definition of $K$ in section 4.3.

---

### Official Review · Reviewer_9AcT · 2023-10-28

**Soundness:** 1 poor
**Presentation:** 1 poor
**Contribution:** 1 poor
**Rating:** 1
**Confidence:** 5

**Summary:**

This paper tends to enhance the robustness of the GCL with Bayesian non-parametric prototype learning and low-rank transductive classification. The paper is based on prototypical contrastive learning. Both two components are from existing papers. The evaluations demonstrate its robustness to noises.

**Strengths:**

1. The robustness of graph contrastive learning is an interesting problem.
2. The experiments basically demonstrate its robustness.

**Weaknesses:**

1. The writing and organization are too poor to get the contribution. Most parts are seriously mixed, including preliminary, motivation, notations, and methods. For example, the second paragraph in section 4.3 should be placed in preliminaries.
2. Many key component details are missing, such as the Bayesian non-parametric prototype learning.
3. The citation is not correct, such as (Zhang & Chen, 2018) in section 4.2 is for link prediction.
4. The novelty is very limited. Low-rank-based transductive classification is not novel, such as [1]. The Bayesian non-parametric prototype learning is also investigated.
5. Some statement is not serious, including the problem description.
6. Some sections are confusing. The title of section 4.1 is RGCL: ROBUST GRAPH CONTRASTIVE LEARNING WITH BAYESIAN NONPARAMETRIC PROTOTYPE LEARNING (BPL). However, the BAYESIAN NONPARAMETRIC PROTOTYPE LEARNING is in section 4.2.
7. Figure 1 needs more explanations in the main body.

[1] 	Andrew B. Goldberg, Xiaojin Zhu, Ben Recht, Jun-Ming Xu, Robert D. Nowak: Transduction with Matrix Completion: Three Birds with One Stone. NIPS 2010: 757-765

**Questions:**

See weaknesses.

---

### Official Review · Reviewer_FuBV · 2023-10-31

**Soundness:** 3 good
**Presentation:** 2 fair
**Contribution:** 2 fair
**Rating:** 6
**Confidence:** 4

**Summary:**

The paper introduces a node classification method for noisy graphs (noise may be in the labels or the node attributes). The proposed LR-RGCL first learns robust node embeddings using Bayesian Prototype Learning for estimating clusters and using contrastive learning. It then uses the learned embeddings for low-rank transductive node classification. On comparing with competing baselines, LR-RGCL performs well on multiple benchmark graph datasets and across multiple noise levels.

**Strengths:**

1. The problem of noisy node classification addressed in this paper is an important one that has been studied in the community for a long time and has significant value.
2. The proposed LR-RGCL relies on contrastive learning which is a well-founded field of study.
3. The experiments included in the paper are extensive and they show that RL-RGCL consistent improves the noisy node classification performance compared to several baselines on multiple datasets across all noise levels.
4. The overall presentation of the paper is good.

**Weaknesses:**

1. The paper lacked a comprehensive section covering works related to graph contrastive learning.
2. The idea of using clustering/cluster labels in GCL is not completely novel. See [1, 2] but I believe that it is still used differently in the proposed work. Even so, it would be nice to compare the proposed method with these, especially CGC[1].
3. Similarly, the idea of using cluster labels as pseudo labels has been explored [3, 4].
4. Clarification on a few things would be useful (see Questions).

All that being said, although the individual components are not necessarily novel, there is still some novelty and effectiveness (as shown by experiment results) in combining them in the way this paper does for the given task. The comment is to potentially include comparisons of the proposed work with these similar ideas.

&nbsp;
&nbsp;
[1] Park, Namyong, et al. "Cgc: Contrastive graph clustering for community detection and tracking." Proceedings of the ACM Web Conference 2022. 2022.

[2] Zhao, Han, et al. "Graph Debiased Contrastive Learning with Joint Representation Clustering." IJCAI. 2021.

[3] Yang, Fan, et al. "Class-aware contrastive semi-supervised learning." Proceedings of the IEEE/CVF Conference on Computer Vision and Pattern Recognition. 2022.

[4] Chakraborty, Souradip, Aritra Roy Gosthipaty, and Sayak Paul. "G-SimCLR: Self-supervised contrastive learning with guided projection via pseudo labelling." 2020 International Conference on Data Mining Workshops (ICDMW). IEEE, 2020.

**Questions:**

1. No related works on GCL/prototypical CL?
2. What is the effect of the estimated $K$? Does the clustering really need to be non-parametric? How does this compare with simple clustering methods like k-means? Any reason they were not included as baselines?
3. For a particular epoch, the $\gamma_k$ is the same irrespective of $k$? So how are the nodes assigned to clusters? Are only the nodes with $z_i = k$ considered for identifying the confident nodes within that cluster?
7. How does the proposed method compare with CGC [1]?
8. How does an ablation study perform based on $L_{proto}$ vs $L_{node}$? What is the significance of each of the components?

&nbsp;
[1] Park, Namyong, et al. "Cgc: Contrastive graph clustering forcommunity detection and tracking." Proceedings of the ACM Web Conference 2022. 2022.

---

### Official Review · Reviewer_X4pk · 2023-11-01

**Soundness:** 2 fair
**Presentation:** 3 good
**Contribution:** 1 poor
**Rating:** 3
**Confidence:** 4

**Summary:**

The paper studies the problem of graph classification with noisy features and labels. The proposed methods are compared with a few baselines.

**Strengths:**

1. The preliminary knowledge has been clearly explained.
2. The experiments are sufficient, to some extent.

**Weaknesses:**

1. The noisy classification problem is not interesting since there have been many studies about the noisy classification of images and tabular datasets. The problem considered in the paper is not related to the graph structure. For graph data, the noisy adjacency matrix would be more worthy of study.

2. The paper is poorly organized. For instance, the connection between Section 4.3 and Sections 4.1 and 4.2 is not clear. It used four pages to introduce the background and related work but gave much fewer details about the proposed methods and experimental setting.

3. The low-rank technique is a very classical method for handling noise. Introducing low-rank factorization to the feature matrix H does not make a significant contribution.

4. The optimization details about the low-rank factorization is missing. The factorization is not a continuous operation and will break off the backpropagation.

**Questions:**

1. What is the connection between (2) and (1)?

2. What is the impact of $r$ in (3)? Is $1-\eta \widehat{\lambda} _r^2$ always nonnegative? According the analysis $\mathcal{U} _{\text {test }}(t) \leq c_1 c_3 r\left(\frac{1}{u}+\frac{1}{m}\right)+\frac{c _2 x}{u}+\varepsilon(t)$, when $r$ is larger, the bound is looser. But in practice, when $r$ is too small, the prediction will not be accurate.

3. The definition of noise level is missing.